# Lung Macrophage Functional Properties in Chronic Obstructive Pulmonary Disease

**DOI:** 10.3390/ijms21030853

**Published:** 2020-01-28

**Authors:** Kentaro Akata, Stephan F. van Eeden

**Affiliations:** 1Department of Medicine, University of British Columbia, Vancouver, BC V6Z1Y6, Canada; kakaredish0421@gmail.com; 2Centre for Heart Lung Innovation, University of British Columbia, Vancouver, BC V6Z1Y6, Canada

**Keywords:** lung macrophages, function, COPD

## Abstract

Chronic obstructive pulmonary disease (COPD) is caused by the chronic exposure of the lungs to toxic particles and gases. These exposures initiate a persistent innate and adaptive immune inflammatory response in the airways and lung tissues. Lung macrophages (LMs) are key innate immune effector cells that identify, engulf, and destroy pathogens and process inhaled particles, including cigarette smoke and particulate matter (PM), the main environmental triggers for COPD. The number of LMs in lung tissues and airspaces is increased in COPD, suggesting a potential key role for LMs in initiating and perpetuating the chronic inflammatory response that underpins the progressive nature of COPD. The purpose of this brief review is to discuss the origins of LMs, their functional properties (chemotaxis, recruitment, mediator production, phagocytosis and apoptosis) and changes in these properties due to exposure to cigarette smoke, ambient particulate and pathogens, as well as their persistent altered functional properties in subjects with established COPD. We also explore the potential to therapeutically modulate and restore LMs functional properties, to improve impaired immune system, prevent the progression of lung tissue destruction, and improve both morbidity and mortality related to COPD.

## 1. Introduction

Chronic obstructive pulmonary disease (COPD) is an inflammatory disease of the lung that involves the lung parenchyma and the airways. COPD is the fourth leading cause of death in the United States and mortality rates of COPD are still rising compared to other chronic diseases [1]. COPD is caused by the continuing inhalation of toxic substances, predominantly cigarette smoke (CS), that are actively or passively inhaled [2]. In contrast to the developed world, in the developing world, exposure to ambient particulate matter (PM) significantly contributes to the burden of COPD [3]. These exposures initiate a persistent innate and, subsequently, an adaptive immune response in the lung. This response induces an impaired tissue repair and a remodeling process characterized by an overproduction of mucus in the central airways, destruction and fibrosis of the small airways, and destruction of the lung parenchyma [4]. The chronic inflammatory process further increases during acute exacerbations and is associated with a poor long-term outcome [5]. In addition, this chronic inflammatory response in the lung is associated with a systemic inflammatory response that contributes to the morbidity and mortality of the disease [6].

## 2. Lung Macrophages (LMs) in Chronic Obstructive Pulmonary Disease (COPD) 

The LMs are key innate immune effector cells that identify, engulf, and destroy pathogens and process inhaled particles, including CS and PM, the main environmental triggers for COPD [7]. The number of macrophages in bronchial alveolar lavage fluid (BAL) is increased in smokers with and without COPD compared with non-smokers [8,9]. This suggests a potential key role for LMs in the chronic inflammatory process in CS-induced lung disease. The reason for this increase in the number of macrophages in the airspace in COPD is unclear. Traves and co-workers showed an increase in the recruitment of blood monocyte-derived macrophages (MDMs) into the airspaces of COPD [10]. Other potential reasons could be prolonged survival of macrophage or increased proliferation of resident macrophages induced by the chronic inflammatory milieu in the lungs [11]. Furthermore, in the inflammatory milieu in the lung, in non-polarized macrophages widely distributed in both small airways and airspaces, some migrate to small airways and M1 macrophages are formed, and others migrate to airspaces and M2 macrophages are formed [12]. The M1 phenotype is generally more pro-inflammatory and is designed to fight off microbes, whereas the M2 phenotype is more anti-inflammatory and assists in tissue remodeling and repair. We suspect that these functions may be altered in chronic inflammatory lung conditions such as COPD. Collectively, these changes suggest that LMs have a pivotal role in both the initiation and progression of the chronic inflammatory process in lung tissues in COPD and this review will focus on the role LMs play in the chronic inflammatory response in the lungs of subjects with COPD. 

## 3. Origins of LMs

Numerous studies in rodents (predominantly mice) showed that LMs are embryonic derived and originate from the yolk sac [13,14,15,16,17], the fetal liver [14,18] and the bone marrow [19,20]. Serena and co-workers showed that the lung is populated with macrophages in three successive waves [21]. The first wave starts at embryonic day 10.5 (E10.5) in yolk sac. F4/80 (which is the major macrophage marker) lineage macrophages from yolk sac spread uniformly throughout the lung interstitium during embryonic development to become the “primitive” interstitial macrophages. The second wave starts 2 days later in the fetal live. Embryonic Mac2 macrophages distribute throughout the lung interstitium and enter the alveoli spaces during the first week of postnatal life, and they remain and sustain themselves as alveolar macrophages with self-renewing potential. The third wave starts from birth during the first week of postnatal life as bone marrow-derived macrophages to establish a new population of “definitive” interstitial macrophage, replacing the “primitive” interstitial macrophages in the lung parenchyma. Airspace macrophages derive from the embryonic precursors that reside in the luminal niche after expansion of the lungs at birth. Both the migration through epithelial cells and engraftment of the precursors into the airway are dependent on L-plastin expression [22], and their differentiation into macrophages in the air spaces is controlled by the expression of granulocyte–macrophage colony-stimulating factor (GM-CSF) [23]. Therefore, the lungs are populated with these embryonic macrophages that are long lived and have the ability of self-renewal, making them important guardians of pulmonary homeostasis. 

## 4. Normal Functions of LMs

LMs are primary immune sentinels, critical in regulating the balance between immune cell defense against pathogens/toxins and tolerance toward innocuous inhaled stimuli to protect lung tissue integrity and to allow effective exchange of CO_2_ and O_2_ transfer [24]. Airspace macrophages patrol the airspaces to keep the airways clear of inhaled particles/pathogens and toxins via phagocytosis and initiating innate immune responses. The resident macrophage pool is then replaced by recruited MDM precursor cells that are recruited into the airway space via transepithelial migration and differentiate into airspace macrophages. The balance between the resident and recruited macrophages is tightly regulated by endogenous inflammatory mediators during the inflammatory response in the lung, triggered by exogenous inhaled stimuli such as particles/pathogens. The complex microenvironment in the lung triggered by inhalation exposures alters the phenotype and function of LMs through modifying their gene transcription [25]. Exposure to interferon-γ (IFN-γ) promotes the production of M1 macrophages. The “classical” activated M1 macrophages have enhanced phagocytic capabilities and antigen-presenting properties similar to T helper (Th1) cells. The produced Th1 cytokines such as interleukin-1β (IL-1β), IL-6, and IL-12 and tumor necrosis factor-α (TNF-α) are all important for clearance of intracellular and bacterial pathogens [26]. M2 macrophages are characterized by the expression of distinguishable surface markers such as mannose receptor (CD206), biosynthetic enzymes such as arachidonate 12,15 lipoxygenases, and other proteins such as chitinases and chitinase-like proteins and matrix metalloproteinase (MMP)-12 [26]. Differentiation into M2 macrophages is driven predominantly by IL-4 and IL-13. However, these macrophages also exhibited pro-inflammatory functions (especially during allergic inflammation) and were further subdivided into M2a–d (M2a promoted by interleukin IL-4 and IL-13, M2b by immune complexes and lipopolysaccharides (LPSs), M2c by glucocorticoids and transforming growth factor beta (TGF-β), and M2d by IL-6 and adenosines [27,28]). Therefore, the microenvironment in the lung has the ability to shape and reshape LMs differentiation dependent on their microenvironment, highlighting the underlying strong plasticity of lung macrophages [29], whether they are resident macrophages or recruited macrophages. This fine tuning of macrophage phenotypes depends on microenvironmental stimuli to maintain a delicate balance between excessive inflammation in response to innocuous antigens and tissue repair and homeostasis (Figure 1). 

## 5. Alveolar Macrophages (AMs) vs. Interstitial Macrophages (IMs)

Lung macrophages mainly consist of airspace (AMs) and interstitial macrophages (IMs) based on their anatomical location. There are less informative studies using IMs because fresh lung tissue is needed to extract these macrophages for functional studies. Subsequent culture of these extracted macrophages could also alter their functional properties. IMs are smaller and morphologically more similar to blood monocytes than AMs [30,31,32]. IMs also have a higher nuclear/cytoplasm ratio, their cytosol includes more vacuoles [33] and they are more heterogeneous in shape compared to AMs [30,31]. 

IM have a lower phagocytic activity of *Saccharomyces cerevisiae* [31] but their phagocytic activity of latex beads is similar to AMs [30], which means lower Fcγ-dependent phagocytic activity in IMs than AMs. IMs expressed more MHC-II (HLA-DR) to function as antigen-presenting cells [30,34]. In the steady state, IMs secrete pro-inflammatory cytokines (IL-1, IL-6 [30], and TNF-α [30,35,36,37]) and anti-inflammatory cytokine (IL-10 [38,39,40,41]). The amount of IL-10 produced by IMs increases in response to stimuli such as LPSs [39], DNA-containing non-methylated CpG motifs (CpG-DNA) [42] or extracts of house dust mite (HDM) [38] and the expression of TNF-α increased in IMs but not in AMs in response to IFN-γ and LPSs [43]. The expression of matrix metalloproteinases (MMPs) in IMs was higher than in AMs [34]. Blood monocytes can transition into IMs that can be recruited into airspaces and transition to AMs via a process of maturation [44,45]. The turnover rate of IMs is shorter in steady state than that of AMs, predominantly regulated by apoptosis [43], resulting in AMs living longer than IMs [46,47,48]. In addition, lung inflammation move monocytes from the bloodstream via lung tissue into airspaces where they differentiate into AMs [49]. Furthermore, macrophages with the AM phenotype have been identified in the lung interstitium [50] and vice versa [42], suggesting that these two populations of macrophages replenish each other dependent on need. This plasticity of LMs complicates studies on LMs [49]. Further in vivo studies are needed to better clarify phenotypic and functional characteristics of LMs and their potential role in the pathogenesis of COPD. 

## 6. Functional Alterations in LMs of COPD

Several studies have shown an increase in the number of macrophages in both sputum and BAL fluid of patients with COPD [51,52,53,54]. These LMs are predominantly from an increased recruitment of blood monocytes from the circulation in response to the monocyte-selective chemokines such as chemokine (C-C motif) ligand 2 (CCL2) and chemokine (C-X-C motif) ligand 1 (CXCL1) [10]. The inflammatory milieu in the lung tissues and airspaces transform these monocytes to be more “macrophage-like,” with the cytokines and chemokines that are characteristic of macrophages [55,56]. The activation of these LMs by stimuli such as CS release inflammatory mediators, including tumor necrosis factor alpha (TNF-α), CXCL1, CXCL8, CXCL6, CCL2, leukotriene B4 (LTB4), and reactive oxygen species (ROS). These mediators are particular efficient in recruiting other innate immune cells such as neutrophils which process and remove pathogens/micro-organisms from the inflammatory focus. When stimulated, LMs also produce and secrete a variety of elastolytic enzymes (including MMP-2, 9, and 12 [57] and cathepsins K, L, and S), which contribute to the intra-cellular and extra-cellular killing and processing of pathogens [58]. LM also have a critical important role as “janitors” in cleaning up or resolving the inflammatory reaction—for example, in removing neutrophils and their products such as elastase from the inflammatory niche [59]. Lastly, LMs have a central role in initiating the adaptive immune response by serving as antigen-presenting cells to lymphocytes [7]. Collectively, LMs play a key part in orchestrating the chronic inflammatory response in lung tissue and airspaces in patients with COPD [11]. To support this concept, Di Stefano and co-workers showed a strong correlation between macrophage numbers in the airways and COPD severity [60] and Finkelstein and co-workers showed increased numbers of macrophages localized in regions of alveolar wall destruction/emphysema [61]. 

### 6.1. Chemotaxis 

Chemotaxis is the movement of cells directed by a chemical stimulus called chemokines [62]. Chemokines are major mediators produced and secreted by macrophages. The chemokine family contains 50 members that are classified into four subfamilies including CC, CXC, C, and CX3C and their receptors consist of CXCR1, 2, 3, 4, and 5 (bind CXC), CCR1 through CCR9 (bind CC), CR1 (binds C), and CX3CR1 (binds CX3C) [62,63]. Chemotaxis induced by chemokines (produced by resident LMs and epithelial cells) are crucial in the recruitment of monocytes from the circulation into lung tissues and airspaces in COPD [64]. The triggers that stimulate the production of these chemokines are inhaled particles and gasses from CS and ambient air pollution [65]. 

Peripheral blood mononuclear cells (PBMCs) migrated towards CXCL9, CXCL10, CXCL11, and CCL5 (RANTES) in non-smokers, smokers and COPD patients. However, the migratory responses of PBMCs from COPD patients were higher than control groups for all these chemokines. This enhanced migratory response PBMCs in COPD is not due to increased receptor numbers on the surface of PBMCs in COPD [66]. Alternatively, this increased migratory response could be due to higher levels of chemokines or the enhanced chemotactic responsiveness of these cells in COPD. PBMCs include other mononuclear cells such as T cells, B cells, and natural killer (NK) cells as well as dendritic cells, which could potentially influence these findings of increased chemotaxis. Ravi and co-workers showed that purified blood monocytes form COPD subjects have demonstrated decreased migratory ability notwithstanding the over-expression of chemotactic receptors such as CCR5 [67]. Together, these findings suggest that the increased numbers of lung macrophage in COPD are not from excessive recruitment of blood monocyte but rather attenuated apoptosis of LMs or an increase in the self-renewal of resident macrophages. Further research is needed to clarify these issues. 

Xuan and co-workers reported that M1 macrophages respond to a wide variety of chemokines including CCL19, CCL21, CCL24, CCL25, CXCL8, CXCL10, and XCL2. CCL7-induced chemotaxis of both M1 and M2 macrophages and its receptor CCR7, which specifically bind CCL19/CCL21, was also detected in both M1 and M2 macrophages. CCR7 was more highly expressed on M1 macrophages than M2 macrophages and its expression was associated with the activation of the ERK and AKT pathways in M1 cells and CCL19/CCL21 responses in M2 macrophages [68]. These findings highlight the differences in functional responses in different macrophage phenotypes. Monocyte chemoattractant protein-1 (MCP-1) is known as CCL2. CCL2 is a CC chemokine and its specific receptor is CCR2, which is expressed by monocytes, macrophages, and T lymphocytes [69,70,71,72] and plays a key role in the migration of monocytes and macrophages. BAL fluid from smokers with or without chronic bronchitis contained higher concentrations of CCL-2 (MCP-1) compared with non-smokers [73], suggesting that this chemokine is a pivotal player in LMs responses, specifically chemotaxis, in COPD [74]. 

### 6.2. Mediator Production by LMs

Studies from several investigators including from our own laboratory have shown that LMs are pivotal for processing inhaled toxins and particles (such as CS and PM) and dictate local lung and systemic inflammatory responses induced by these exposures [7]. When exposed, they release mediators that dictate and characterize both the innate and adaptive immune response that impact the local lung inflammatory response and could determine the ultimate development and progression of COPD [75]. These inhaled stimuli promote the production of pro-inflammatory cytokines such as IL-6, IL-1β and TNF-α and in vivo models in human and rodents [76,77,78,79] have shown that LMs secrete more pro-inflammatory cytokines than airway epithelial cells [64,65]. In addition, LMs and airway epithelial cells coordinate the local inflammatory reaction due to their close proximity [64,80].

We have also shown that mediators released from LMs following CS and ambient PM exposure promote the release of monocytes from the bone marrow [81], which are recruited into the lungs where they differentiate into tissue and airspace macrophages, replenishing resident LMs [64]. These LMs are recruited from circulating monocytes pool [82]. The turnover of monocytes in the bone marrow is stimulated by macrophage colony-stimulating factor (M-CSF), GM-CSF, and IL-6 and also induces their release into the circulation [83,84]. IL-6 activates the formation of a monocytic colony in hematopoietic progenitor cells in concert with GM-CSF [85]. Mediators such as IL-1β, IL-6, GM-CSF, and TNF-α induce the production of MCP-1, which is a major contributor to the recruitment and replenishment of peripheral blood monocytes into the alveolar space [86]. The stimulation of the bone marrow is related to the ability of PM phagocytosis by AMs [84]. In addition to cytokines and chemokines released by LMs, they also produce and release proteases such as MMPs that have the potential to damage tissues and amplify the inflammatory response, contributing to the tissue destruction seen in COPD lung tissues. Russel and co-workers showed that LMs from patients with COPD increased their release of MMP-9 [52] and release significantly less of the tissue inhibitor of metalloproteinase (TIMP)-1 (an important anti-elastolytic molecule) than non-smokers [52,87]. This imbalance between protease (MMP) and anti-protease in the lungs has also been demostrated by other investigators [88,89,90,91]. MMPs are known to degrade the epithelial basement membrane, leading to the induction of airway epithelial apoptosis [92,93]. 

Collectively, mediators produced and release by LMs when exposed to CS and PM, classical triggers for the development of COPD, depend on inhaled particles/pathogens as well as host factors such as the innate and acquired immune responses in the lung. Further studies are needed to clarify the pathogenesis of COPD considering these factors.

### 6.3. Phagocytosis of LMs

There are several lines of evidence that both phagocytosis and efferocytosis in LMs from COPD subjects are impaired. Phagocytosis is defined as the engulfment of PM of >0.5–1 µm by professional and/or non-professional phagocytic cell types [94,95]. Efferocytosis is the process of engulfment of dying and dead cells and their cellular products by phagocytes [96,97] and includes the uptake of solutes via macropinocytosis [98,99,100]. Professional phagocytes are composed of macrophages, immature dendritic cells (DCs), tissue-infiltrating monocytes, neutrophils, and eosinophils. Non-professional phagocytes in the lung consist predominantly of epithelial cells but include cells such as connective tissue cells, muscle cells and endothelial cells [101,102]. Removing cell debris and eliminating cells undergoing programmed cell death are crucial processes in containing ongoing inflammation and the resolution of inflammation [103,104]. LMs from COPD patients and CS-treated animals have decreased ingested apoptotic cells [105] and microorganisms, such as *Haemophilus influenzae* [24,106,107,108,109], *Candida albicans* [110,111], *Escherichia coli*, *Moraxella catarrhalis* [106,107], and *Streptococcus pneumoniae* [24,106,112], apoptotic neutrophils [113,114], eosinophils [115], and airway epithelial cells [116,117,118]. Interestingly, LMs from COPD patients have similar phagocytosic activities of inert beads than cells from control subjects [109,118], indicatings that CS does not cause a generalized impairment in phagocytic capacity of LMs. The reason for this specific phagocytic impairment in COPD is unclear. Droeman and co-workers suggest that the low expression of Toll-like receptors (TLR)2 on LMs from COPD subjects could contribute to this phagocytic defect in COPD. TLRs are recognition molecules for a variety of pathogens, including bacteria, viruses, fungi, and parasites, with TLR2 as a pivotal molecule of the initial step in a cascade of events leading to an innate immune response and the subsequent development of adaptive immunity to pathogens [119]. Todt and co-workers also showed low levels of TLR3 on LMs [120], Hodge and colleagues showed low levels of efferocytic recepters such as CD31, CD44, and CD91 on LMs [117,121,122] and Pons and colleagues show a lower expression of human leukocyte antigen (HLA) class II and CD80 in LMs from patients with COPD than smokers and non-smokers [123]. One of the potential mechanisms of this low level of receptors involved in recognizing particles and pathogens in COPD is high levels of oxidant stress and proteolytic reactions that reduce efferocytic opsonins and cleave efferocytic receptors, leading to the accumulation of apoptotic cells and debri in lung tissues from COPD patients. These apoptotic cells that are not eliminated undergo secondary necrosis, which stimulates NKT cells, activates DCs, and drives the maturation of T cells, with the further release of inflammatory mediators that decrease efferocytosis, causing a vicious cycle [124]. These studies suggest that the phenotypic characteristics of LMs in COPD significantly impact their functional responses, in particular their ability to phagocytose particular matter including pathogens. 

## 7. Apoptosis in Lung

Several lines of evidence support the notion that enhanced apoptosis of structural lung cells in the lung might possibly be an important upstream event in the pathogenesis of COPD [92,125]. The structural cells involved are predominantly airway and alveolar epithelial and endothelial cells. Defective removal of these dead or dying cells is postulated to contribute to the ongoing inflammatory response in the COPD, even in subjects that have stopped smoking [126]. Apoptosis or programmed cell death allows the removal of damaged or dying cells. Cells are pushed into becoming apoptotic due to a collapse of cellular homeostasis (intrinsic pathways) or due to noxious extracellular stimuli (external pathway) [127]. Intrinsic factors triggering apoptosis occur as a result of growth factor withdrawal, oxidative stress, endoplasmic reticular stress, and DNA damage. In the lung, oxidative stress occurs as a consequence of inhaled CS [128] with ROS and ROS inducers [129] reaching the nucleus to cause oxidative DNA damage and eventually cell death [130]. The transcription factor nuclear factor erythroid 2-related factor 2 (NRF2; also known as NFE2L2) is important to regulate antioxidant and cytoprotective genes stimulated by oxidative stress. Mitochondria release cytochrome C (Cyt C) induced by physical and chemical stress combined with apoptotic protease activating factor-1 (Apaf-1) and procaspase-9, leading to form the apoptosome complex [131,132,133], resulting in the activation of caspase-9 and caspase-3 as the final executor of apoptosis [134]. An enhanced oxidative stress environment in the COPD promotes this intrinsic apoptotic pathway. Furthermore, in response to extracellular stimuli, death factors (TNF-α and Fas ligand (FasL)) initiate apoptosis by binding to death receptors (tumor necrosis factor receptor I (TNFRI) and Fas, which is expressed by alveolar and bronchial epithelial cells, club cells, macrophages, and myofibroblast in the lung [127]. This results in the multimerization of the death receptor and the formation of the Fas-associated death domain (FADD) including the death-inducing signaling complex. This FADD interacts with initiator caspase-8 [135,136], leading to the autolytic activation of procaspase-8 to caspase-8, which releases caspase-activated DNAse (CAD) from its inhibitor (ICAD), inducing DNA fragmentation [137]. 

## 8. LMs and Apoptosis in COPD

LMs are key effector cells that identify, engulf, and destroy pathogens and process inhaled particles from CS and ambient PM exposure, the main environmental triggers for COPD. The chronicity of these exposures is associated with persistent lung tissue and airway inflammation and repair, which creates a vicious cycle of tissue damage and repair, which becomes dysfunctional over time [138,139]. LMs play a crucial part in this repair of damage and inflamed tissues by processing and removing dead and dying cells through phagocytosis. The chronic exposure of LMs to inhaled PM and CS, as well as pathogens and their toxic products (LPSs), in addition to their overstretched “janitorial” functions, eventually promotes apoptosis in macrophages themselves [140,141,142] (Figure 2). Ultimately, this aberrant inflammatory response and dysfunctional repair process may lead to the development and/or progression of COPD. 

### 8.1. LM Apotosis Accociated with Cigarette Smoke Exposure 

The vitro studies in mice, rats, human LMs, and human blood MDMs showed that CS exposure leads to apoptosis in these macrophages [142]. The apoptosis induced in these models was related to an increase in oxidative stress, Bax protein accumulation, mitochondrial dysfunction, and mitochondrial cytochrome c release, but was unrelated to p53, Fas, and caspase activation [142]. CS extract (CSE) led to apoptosis at a lower density (10% to 25%) but to cell death and necrosis at higher density (50% to 100%) [143], confirmed by earlier in vitro studies [144,145]. Machiya and co-workers reported that CS-induced apoptosis in LM was partly dependent on caspase-3, in contrast to the reports of Aoshiba [142]. The involvement of caspase-3 in CS-induced apoptosis was supported by two findings: (1) the inhibitor of caspase-3 (ZDEVD-FMK) reduced apoptosis by CSE; (2) caspase-3 was activated by CSE in the macrophage cell line in a dose-dependent manner [140]. These studies have never been confirmed in human macrophage. In contrast to these studies, oxidative stress induced cytoplasmic p21^CIP1/WAF1^ and Bcl-xL (anti-apoptotic proteins) which were elevated in LMs from smokers in contrast to nonsmokers [141]. Furthermore, RT-PCR analyses for BAL cells revealed MafB mRNA expression (an inhibitor of apoptosis) was elevated in 6-month-old CS-exposed mice compared with those in control mice [140]. These reports showed that anti-apoptotic factors are induced in macrophages, although cell apoptosis was not quantified in either of these studies. Collectively, these studies suggest that the inflammatory response induced by stimuli such as CS and PM exposure generates anti-apoptotic signals in LMs, potentially as a protective response to these exposures. However, due to the intensity and chronicity of these exposures and the chronic inflammatory response they induced, LMs will enventually be more apoptotic and that could impact their functional properties such as phagocytosis. Further studies are needed to determine the contribution of enhanced apoptosis of LMs to their defective functional properties. 

### 8.2. LM Apoptosis Associated with Pathogen Exposure

Lung and MDMs from COPD patients chronically colonized by bacteria [146], such as H. influenzae, S. pneumoniae [24], or M. catarrhalis [147], have reduced (>50%) macrophage phagocytic function. These macrophages have a normal capacity to phagocytose latex beads [109,118,148] but not bacterial pathogenes [149]. One of the strategies pathogens use to avoid immune-mediated killing is to induce apoptosis in immune cells such as macrophages. Bacteria has a variety of mechanisms of inducing macrophage apoptosis, which include the activation of caspase-1 [150,151], caspase-3 [152], and Fas [153] or the caspase-independent activation of apoptosis through mitochondrial factors [154], inhibition of nuclear factor kappa-light-chain-enhancer of activated B cells (NF-κB) and TNF-α [155], or signaling via TLRs [156]. For example, chronic colonization of the lower airways with S. pneumoniae induces macrophage apoptosis in a Fas-independent manner via the activation of caspases [157]. The colonization of the lower airways is an important risk factor associated with frequent exacerbations of COPD and also the progression of the disease process in the lungs [158]. Therefore, any future studies on LMs functional properties and in particular apoptosis of macrophages should consider the relationship between colonization of the airways and apoptosis of macrophage. 

## 9. LM Apoptosis vs. Senescence in COPD

Cellular senescence is a state in which cells permanently stop dividing and is strongly linked to aging [159]. Cellular senescence is caused by telomere shortening, cellular stress including oxidative stress, oncogene activation, DNA damage, and chromatin abnormality [160]. Several studies have documented reduced telomere length in smokers that develop COPD [161,162,163,164]. Oxidative stress related to CS exposure is a key driver in cellular senescence in the lung [165,166]. Cells have a limited number of divisions and, with advancing age, once DNA damage occurs, cells can no longer be repaired effectively. They enter a state of cellular senescence and, subsequently, die by apoptosis through p53 and p21^CIP1/WAF1^ and the p16^INK4a^/retinoblastoma protein pathways activated [167]. LMs play a key role in clearing these senescent cells. Progenitor cells are needed to maintain cell numbers and, with aging, this turnover system may become less efficient or exhaust the regenerative capacity of stem cells [167]. If these senescent cells are not removed efficiently, they will accumulate, and because these cells remain metabolically active and have the potential to produce proinflammatory cytokines, chemokines, and MMPs, that could eventually lead to organ dysfunction (termed the senescence-associated secretory phenotype (SASP)) [168]. In the cell-line, RAW264.7 macrophages, have been shown to undergo cell cycle arrest associated with an increase in p21^CIP1^and p53 accumulation in response to hyperoxia (95% O_2_) [169]. This signal is related to cell senescence in macrophages [170,171]. Similarly, oxidative stress due to CS exposure, which is known to induce p53 and p21^CIP1/WAF1^ and p16^INK4a^/retinoblastoma and eventual cell cycle arrest, could lead to apoptosis of lung macrophage. The role of macrophages in clearing senescent cells in the lung, and senescence of macrophages themselves, is poorly understood and further studies are needed to clarify the role of LMs senescence in the pathogenesis of COPD.

## 10. Therapeutic Options to Improve Macrophage Function in COPD

Bronchodilation, anti-inflammatories and anti-infection drugs are generally used as treatment strategies for COPD. These interventions have been shown to improve COPD symptoms and prevent exacerbations, but there is very little evidence that these approaches improve airway remodeling, irreversible obstructive airflow, and a decline in forced expiratory volume in 1 s (FEV_1_) over time. Smoking cessation is the only intervention that has convincingly been shown to prevent the decline in FEV_1_ over time. Changing the behavior of key immune cells (such as macrophages) or pathways involved in the pathogenesis of COPD has the potential to attenuate the chronic inflammatory response and augment the normal repair process of lung tissues, thereby preserving or regaining lung function over time. This section will focus on therapeutic interventions that impact LMs function. 

### 10.1. Macrolides

Macrolides (erythromycin, clarithromycin, and azithromycin) are therapeutic agents beyond antimicrobials. The recent meta-analysis showed that low-dose macrolide therapy significantly reduced exacerbations (OR 0.28 CI 0.12-0.68) [172]. Macrolides are known to have immune-modulatory effects on chronic airway inflammation. They can suppress activator protein 1 (AP1) and the NF-κB mediated cascade, leading to a reduction in the production of pro-inflammatory mediators such as IL-1, IL-6, IL-8, and TNF-α, predominantly made by LMs in COPD patients. Macrolides also impact the recruitment of monocytes into lung tissues and promote monocyte-to-macrophage differentiation, improve macrophage efferocytosis, thus suppressing excessive inflammatory responses related to cell necrosis, change macrophage phenotype to improve bacterial clearance, and raise macrophage cytocidal activity in COPD [121,122,173]. In vitro studies have shown that the 14 member macrolides (clarithromycin and erythromycin) as well as the 15 member macrolides (azithromycin) improved LMs efferocytosis of apoptotic neutrophils through the phosphoserine (PS) pathway, but the 16 member macrolides had no effect [174]. These effects were not related to changes in recognition receptors such as CD31 and CD91 [122]. In addition, azithromycin increased phagocytic ability, which was related to an increase in CD206 expression and involved the PS pathway [121,122]. In COPD patients, azithromycin improved efferocytosis of LMs with increased CD206 expression [121]. In addition, in a mice model, clarithromycin has been shown to prevent progress of emphysema. These findings were associated with a significant reduction in macrophages in lung tissues, suggesting that a reduction in macrophage activity induced by clarithromycin was responsible for this “emphysema-sparing” effect [175]. Apart from their other anti-inflammatory properties, macrolides are promising compounds to target macrophages’ dysfunctional properties and specifically to augment efferocytosis and decreased recruitment. This could attenuate inflammatory responses and augment repair processes in lung tissues of COPD, thereby decreasing disease progression and morbidity. 

### 10.2. Collectins 

The collectin family consists of the mannose-binding lectin (MBL), the surfactant proteins A (SP-A) and D (SP-D), all with a collagen-like amino (N)-terminal regions and C-type (calcium-dependent) carbohydrate-recognition domains (CRDs). SP-A and SP-D are surfactants with host-defence functions, synthesized by alveolar type II cells [176,177], club cells, and submucosal cells [178,179]. SP-A, SP-D, and MBL enhance the uptake of particles and pathogens by three different mechanisms in in vitro and in vivo studies [180]: a) by opsonizing pathogens [181,182], b) by functioning as activation ligands [183], and c) by upregulating the expression of cell-surface receptors that are involved in pathogen recognition [184,185]. Hodge and co-workers showed lower levels of MBL in both current and ex-smoker COPD patients than those in non-smoking controls [121]. They also reported that treatment with MBL via nebulizer improved efferocytosis of both alveolar and tissue macrophages and decreased cell numbers in a smoking mouse model. They showed an MBL-mediated increase in phagocytic ability partially related to the renin–angiotensin system (Ras)-related C3 botulinum toxin substrate (Rac) 1/2/3 signaling pathway, which mediates actin cytoskeleton rearrangement required for macrophage engulfment of apoptotic cells [186]. Collectively, this suggests a potential role for lectin therapy in attenuating the chronic inflammatory response in lung tissue of subjects with COPD (see Figure 4). 

### 10.3. Statins

The 3-hydroxy-3-methylglutaryl coenzyme A (HMG-CoA) reductase inhibitors, or statins, are commonly used as cholesterol-lowering drugs and reduce the blood concentration of inflammatory cytokines in addition to lowering cholesterol levels in the blood [187,188,189]. They also have anti-inflammatory, anti-oxidant, anti-thrombogenic, and vascular function-restoring actions [190] and decrease mortality from cardiovascular disease [191]. Alexeeff and co-workers showed statin use in patients with COPD results in a reduced decline in FEV_1_ over time compared to non-users (FEV_1_ of 23.9 mL/year in non-users vs. 10.9 mL/year in statin users) [188]. Additionally, statins have been shown to reduce mortality in COPD patients [192,193]. The postulated reasons for this beneficial effect of statins in COPD are their anti-inflammatory properties [194,195,196], their antioxidant capacity [197], and their potentiation of efferocytosis in LMs [198]. In LMs, the regulation of efferocytosis is controlled by the Ras homolog family member (Rho) family guanosine triphosphatases (GTPases) such as RhoA [99,199,200] and Rac-1/cell division control protein 42 homolog (CDC42)/RhoG [201,202]. Statins regulate prenylation of Rho-GTPases, with blocking of HMG-CoA reductase decreasing the production of mevalonate and downstream prenylation substrates such as farnesyl pyrophosphate (FPP) and geranylgeranyl pyrophosphate (GGPP) (Figure 3). Lovastatin increased efferocytosis in the murine lung and ex vivo LMs from COPD patients in a HMG-CoA reductase-dependent manner. These findings show that statins enhance efferocytosis via RhoA inhibition in vitro and in vivo [198]. CS and other oxidant stressors, suppress efferocytosis via activating RhoA [203], and antioxidant therapy reduces these effects [197]. In addition, statins increase the production of peroxisome proliferator-activated receptor gamma (PPARγ), which also increases efferocytosis of LMs [204]. CS exposure induces an increase in the production of MMP-9 in LMs isolated from COPD patients compared to those from normal volunteers or healthy smokers [52]. The statin, Lovastatin, suppresses MMP-9 from CSE mouse peritoneal macrophages (MPMs) and human monocyte-derived macrophages (HMs) [205]. This reduction in CSE-induced MMP-9 production in LMs is mediated via inhibition of RAS activation, subsequently downstream Raf-MEK, ERK or PI3K-Akt activation, and AP-1 and NF-κB stimulation [206]. Together, these studies demonstrate the anti-inflammatory properties of statins and the potential benefit in attenuating the chronic LM-mediated inflammatory response in COPD (Figure 4). 

### 10.4. Phosphodiesterase (PDE) Inhibitors

The PDE inhibitors have both bronchodilator and anti-inflammatory affects via inhibition of PDE4 and histone deacetylase-2 (HDAC2) activation [207]. The PDE4 enzyme family hydrolyze cAMP and are expressed in a variety of innate immune cells such as macrophages, eosinophils and neutrophils as well as adaptive immune cells such as T-cells and B-cells [208]. The PDE4 inhibitors are predominantly immune modulating and thereby anti-inflammatory by reducing the production of TNF-α from macrophages and attenuating tissue destruction by inhibiting the production of specific matrix metalloproteinases [209]. Selective PDE4 inhibitors are broad-spectrum anti-inflammatory drugs, inhibiting the production of cytokine and chemokine from macrophages. The second generation of PDE4 inhibitors such as roflumilast have clinical benefits that are primarily based on their ability to inhibit macrophage-related chronic lung inflammation in subjects with COPD [210,211]. The beneficial effects of PDE inhibitors on cardiovascular co-morbidities in COPD are thought to occur via their effects on atherosclerotic blood vessel wall disease, including macrophage inflammatory activity [211]. 

## 11. Conclusions

LMs have a central regulatory role in the initiation and progression of lung injury due to the exposure to noxious gases and particulate matter. A consistent finding in COPD is an increase in the number of LMs compared to control non-smokers [8,9]. The reason for this increase is unclear, but it suggests a significant role for these macrophages in the pathogenesis of the chronic inflammation and aberrant repair in COPD. LMs have inherent plasticity, which is largely determined by their microenvironment and is reflected in a variety of different LM phenotypes in COPD. These different LM phenotypes have different functional properties. Functions such as phagocytosis have been shown to be consistently reduced by CS and in COPD, while other functions such as mediator production and apoptosis showed a more variable response in COPD. Further investigation into the distinct role played by the various macrophage subsets in patients with COPD is warranted.

Considering the central role of LMs in COPD, targeting these LMs to attenuate their inflammatory responses and augment their reparative properties is reasonable. Therapeutics such as the new generation of macrolides show promise, while others such as PDE inhibitors, statins and inhibitors of collectins needs further study. 

## Figures and Tables

**Figure 1 ijms-21-00853-f001:**
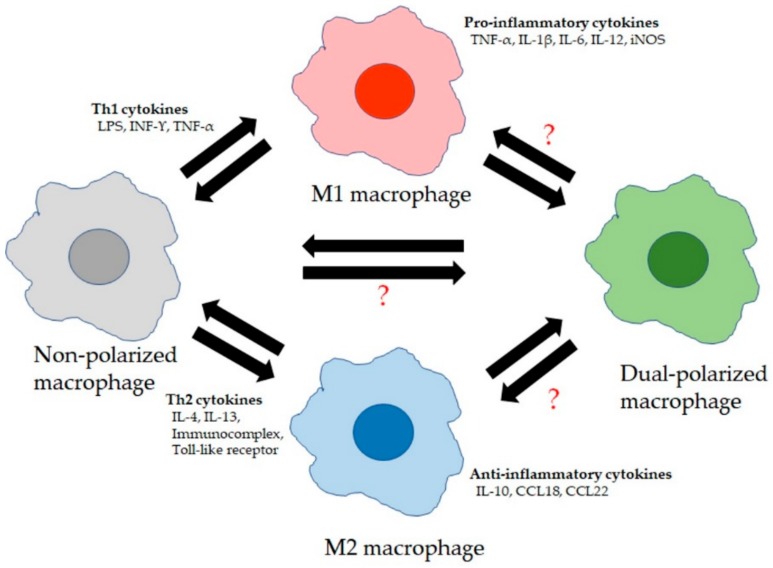
Lung macrophage (LM) phenotypes: LMs consist of at least four broad phenotypes. Non-polarized macrophages are M1 and M2 marker-negative and secrete low levels of cytokines. They become polarized to M1 macrophages due to stimulation by T helper type 1 cell (Th1) cytokines and secrete high levels of pro-inflammatory cytokines. Polarization to M2 macrophages is due to exposure to Th2 cytokines and these M2 macrophages secrete high levels of anti-inflammatory cytokines. “?” The role, functional properties, and forming process of dual-polarized macrophages (M1 and M2 marker-positive) are still unclear.

**Figure 2 ijms-21-00853-f002:**
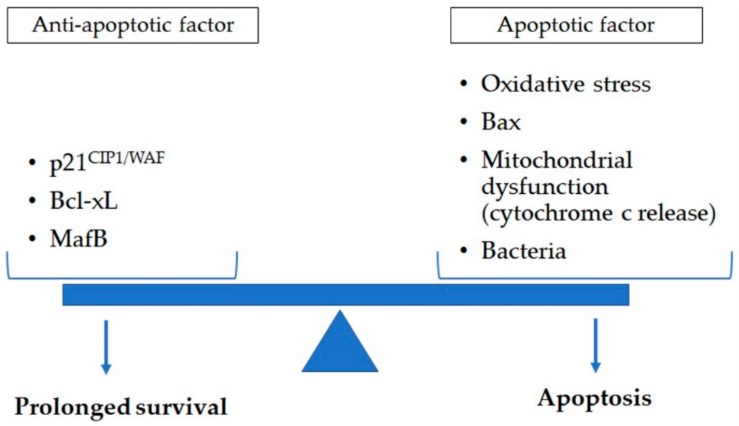
Apoptosis in lung macrophage is determined by a variety of apoptotic (oxidative stress, Bax, and mitochondrial dysfunction such as cytochrome c release) and anti-apoptotic (p21^CIP1/WAF^, Bcl-xL, and MafB) factors. Exposure to cigarette smoke and pathogens impact this balance of anti-apoptotic and apoptotic factors. This could contribute to reduced clearance of dead cell and cell debri augmenting the chronic inflammatory response in lung tissues of chronic obstructive pulmonary disease subjects. Bax: B-cell lymphoma protein 2-associated X; Bcl-xL: B-cell lymphoma-extra large; MafB: v-maf avian musculoaponeurotic fibrosarcoma oncogene homolog B.

**Figure 3 ijms-21-00853-f003:**
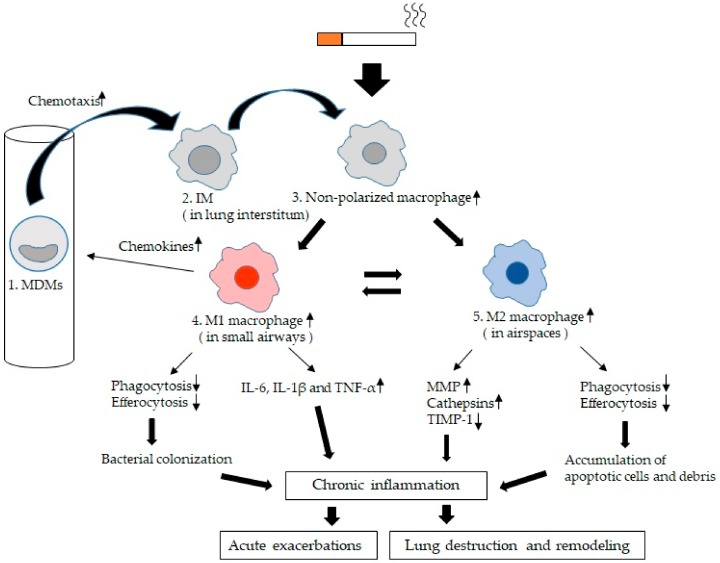
Lung macrophage (LM) kinetics and phenotypes in chronic obstructive pulmonary disease (COPD): 1. Blood monocyte-derived macrophages (MDMs) migrate into small airways and airspaces and this migration is activated by chemokines such as CCL2 or monocyte chemoattractant protein-1 (MIP-1) [69], CXCL9, CXCL10, CXCL11, and CCL5 (RANTES) [66]. 2. Interstitial macrophages (IMs). MDMs are recruited to the lung interstitium and differentiate into IMs [49]. 3. Non-polarized macrophages. IMs recruited into both small airways and larger airspaces are initially non-polarized and are polarized to either M1 macrophages or M2 macrophages dependent on the microenvironment [12]. 4. M1 macrophages. They secrete pro-inflammatory cytokines such as IL-6, IL-1β and TNF-α [76,77,78,79], which could damage lung tissues if secreted in excess. In COPD, phagocytosis and efferocytosis of microorganisms of M1 macrophages is impaired [24,106,108,109,110,111,112], which may lead to bacterial colonization. 5. M2 macrophages or alternative acitivated macrophages are involved in the resolution of inflammation and tissue repair. In COPD, they produce MMPs (MMP-2, 9, and 12) [52,57] and cathepsins (cathepsins K, L, and S) [58] as proteases and release significantly less of the tissue inhibitor of metalloproteinase (TIMP)-1 as a anti-elastolytic molecule [52,87], which could induce tissues damage and amplify the inflammatory response contributing to tissue destruction. Phagocytosis and efferocytosis of cell debris and apoptotic cells are also impaired [105,113,114,115,116,117,118]. These apoptotic cells which are not eliminated undergo secondary necrosis with further release of inflammatory mediators that decrease efferocytosis, causing a vicious cycle [124]. The chronic inflammatory process further increases during acute exacerbations and is associated with a poor long-term outcome [5].

**Figure 4 ijms-21-00853-f004:**
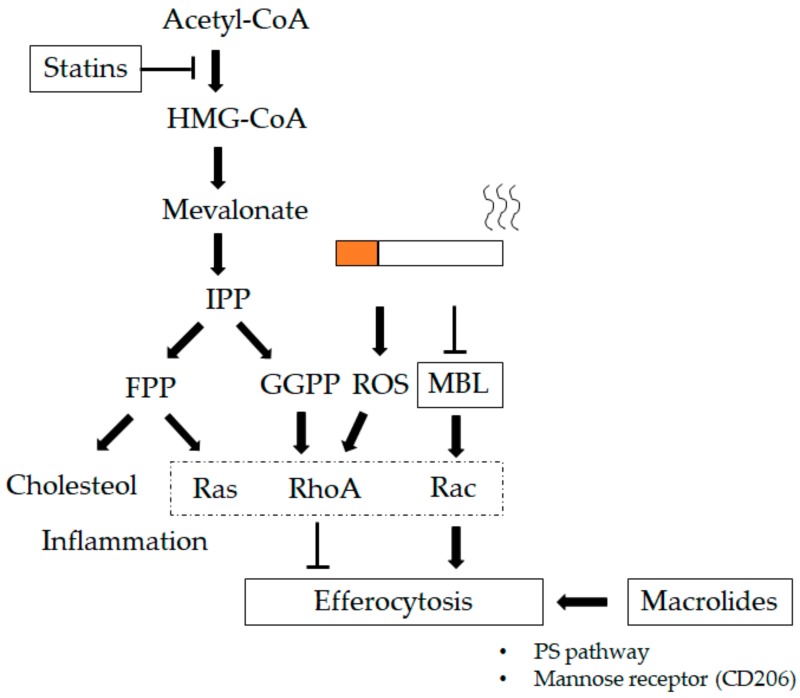
Therapeutics impacting lung macrophages (LMs). Macrolides increase the phagocytic ability of LMs, which could be related to an increase in mannose receptor expression (CD206) and involve the phosphoserine (PS) pathway [121,122,174]. Statins regulate prenylation of the renin–angiotensin system (Ras) homolog family member (Rho)-guanosine triphosphatases (GTPases) that block 3-hydroxy-3-methylglutaryl coenzyme A (HMG-CoA) reductase, decreasing the production of mevalonate and downstream prenylation substrates such as farnesyl pyrophosphate (FPP) and geranylgeranyl pyrophosphate (GGPP). Statins strengthen efferocytosis mediated by RhoA inhibition through GGPP in addition to lowering cholesterol through FPP [198]. Mannose-binding lectin (MBL) increases the phagocytic ability of LMs via the Ras-related C3 botulinum toxin substrate (Rac) 1/2/3 signaling pathway, which mediates actin cytoskeleton rearrangement required for macrophage engulfment of apoptotic cells [186]. T-bar define blocking activity

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
