# Peer review of "Lung Macrophage Functional Properties in Chronic Obstructive Pulmonary Disease"

_ijms, 2020, doi:10.3390/ijms21030853_

Round 1

Reviewer 1 Report

The paper of Akata and van Eeden summarizes the recent data related to the role of lung macrophage in Chronic Obstructive Pulmonary Disease.

Most of the significant studies from literature are reported and discussed, including the work of the authors.

Overall, the review is well presented, structured and written.

Minor points:

The review is written in an educational manner, thus I think that the authors should explain in better way what does it means E10.5 (in the paragraph Origins of LM) and also that F4/80 is the major macrophage marker.

In the legend of the Figure 2 there are too many parenthesis that make difficult the reading. Consider to put the abbreviations at the end of the paper.

Lane 37, page 8: hyperoxia (95% O2) instead of (95% O2).

Figure 3: the “S” of ROS is under the arrow

Author Response

Responses to the reviewer’s comments (Reviewer 1):

Reviewer #1:

We really appreciate your efforts to review our manuscript and your important review comments. According to your comments, we have revised our manuscript as follows. Minor comments

(1) The review is written in an educational manner, thus I think that the authors should explain in better way what does it means E10.5 (in the paragraph Origins of LM) and also that F4/80 is the major macrophage marker.

Response:

Thank you very much for your important comments. We changed the sentence as below.

Page 2, line 20-21

Before

The first wave starts at E10.5 in yolk sac. F4/80 lineage macrophages …

After

The first wave starts at embryonic day 10.5 (E10.5) in yolk sac. F4/80 (which is the major macrophage marker) lineage macrophages…

(2) In the legend of the Figure 2 there are too many parenthesis that make difficult the reading. Consider to put the abbreviations at the end of the paper.

Response:

We sincerely apologize our description in Figure 2. We have changed it as follow.

Figure 2

Before

Figure 2. Apoptosis in lung macrophage is determined by a variety of apoptoric (oxidative stress, B-cell lymphoma protein 2 (Bcl-2)-associated X (Bax), mitochondrial dysfunction (cytochrome c release) and anti-apoptostic (p21CIP1/WAF, B-cell lymphoma-extra large (Bcl-xL), and v-maf avian musculoaponeurotic fibrosarcoma oncogene homolog B (MafB) factors. Exposure to cigarrete smoke and pathogens impact this balance of anti-apoptotic and apoptotic factors. This could contribute to reduced clearance of dead cell and cell debri’s augmenting the chronic inflammatory response in lung tissues of chronic obstructive pulmonary disease subjects.

After

Figure 2. Apoptosis in lung macrophage is determined by a variety of apoptotic (oxidative stress, Bax, and mitochondrial dysfunction such as cytochrome c release) and anti-apoptostic (p21CIP1/WAF, Bcl-xL, and MafB) factors. Exposure to cigarrete smoke and pathogens impact this balance of anti-apoptotic and apoptotic factors. This could contribute to reduced clearance of dead cell and cell debri’s augmenting the chronic inflammatory response in lung tissues of chronic obstructive pulmonary disease subjects.

Add

Page 13, line 2-5

Abbreviations

Bax    B-cell lymphoma protein 2-associated X

Bcl-xL  B-cell lymphoma-extra large

MafB   v-maf avian musculoaponeurotic fibrosarcoma oncogene homolog B

(3) Lane 37, page 8: hyperoxia (95% O2) instead of (95% O2).

Figure 3: the “S” of ROS is under the arrow.

Response:

We sincerely apologize our description. We have changed it as bellow.

Page 8, line 46

Before

hyperoxia (95% O2)

After

hyperoxia (95% O2)

Figure 3

We have changed the figure.

Responses to the reviewer’s comments (Reviewer 2):

Reviewer #2:

We really appreciate your efforts to review our manuscript and your important review comments. According to your comments, we have revised our manuscript as follows.

Minor comments

The general part of the review occasionally appears unstructured and sometimes redundant. I would suggest to carefully scan Page 1-3 for redundant passages and rather explained it in a structured manner once than in short passages multiple times.

Response:

We have removed redundant and repetitive sections as requested (see MS)

A section about the different types of lung macrophages according to their localization would add a new perspective to this review: alveolar vs interstitial. How are the different? What are the differences? Functions? Phagocytic capacity etc? A small section about their different characteristics would be nice. How are they affected by COPD?

Response:

Thank you very much for your important comments. We also think it is important of the difference between alveolar macrophages and interstitial macrophages. Therefore, we added the paragraph as below.

Page 3, line 22-Page 4, line 10

Alveolar Macrophage (AM) vs Interstitial macrophage (IM)

Lung macrophages mainly consist of airspace (AM) and interstitial macrophages (IM) based on their anatomical location. There are less informative studies using IM because fresh lung tissue is needed to extract these macrophages for functional studies. Subsequent culture of these extracted macrophages could also alter their functional properties. IMs are smaller and morphologically more similar blood monocytes than AM [30-32]. IMs also have a higher nuclear/cytoplasm ratio and their cytosol includes more vacuoles [33] and are more heterogeneous in shape compared to AM [30, 31].

IM have a lower phagocytic activity of Saccharomyces cerevisiae [31] but their phagocytic activity of latex beads is similar to AM [29], which means lower Fcϒ-dependent phagocytic activity in IM than AM. IM expressed more MHC-II (HLA-DR) to function as antigen-presenting cells [30, 34]. In the steady-state, IMs secrete pro-inflammatory cytokines (IL-1, IL-6 [30], and TNF-α [35-37]) and anti-inflammatory cytokine (IL-10 [38-41]). The amount of IL-10 produce by IMs increases in response to the stimuli such as LPS [39], DNA containing non-methylated CpG motifs (CpG-DNA) [42] or extracts of house dust mite (HDM) [38] and expression of TNF-α increased in IMs but not in AMs in response of IFN-ϒ and LPS [43]. Expression of matrix metalloproteinases (MMP) in IM was higher than in AM [34]. Blood monocytes can transition into IMs that can be recruited into airspaces and transition to AMs via a process of maturation [44, 45]. The turnover rate of IMs is shorter in steady-state than that of AMs, predominantly regulated by apoptosis [43] resulting in AMs living longer than IMs [46-48]. In addition, lung inflammation recruited monocyte via lung tissue into airspaces where they differentiation into AM [49]. Furthermore, macrophages with AM phenotype have been identified in the lung interstitium [50] and vice versa [42] suggesting that these two populations of macrophages replenish each other dependent on need. This plasticity of LM complicate studies on LM [49]. Further in in vivo studies are needed to better clarify phenotypic and functional characteristics of LM and their potential role in the pathogenesis of COPD. 

Xue, J.; Schmidt, S. V.; Sander, J.; Draffehn, A.; Krebs, W.; Quester, I.; De Nardo, D.; Gohel, T. D.; Emde, M.; Schmidleithner, L.; Ganesan, H.; Nino-Castro, A.; Mallmann, M. R.; Labzin, L.; Theis, H.; Kraut, M.; Beyer, M.; Latz, E.; Freeman, T. C.; Ulas, T.; Schultze, J. L., Transcriptome-based network analysis reveals a spectrum model of human macrophage activation. Immunity 2014, 40, (2), 274-88. Hoppstadter, J.; Diesel, B.; Zarbock, R.; Breinig, T.; Monz, D.; Koch, M.; Meyerhans, A.; Gortner, L.; Lehr, C. M.; Huwer, H.; Kiemer, A. K., Differential cell reaction upon Toll-like receptor 4 and 9 activation in human alveolar and lung interstitial macrophages. Respiratory research 2010, 11, 124. Fathi, M.; Johansson, A.; Lundborg, M.; Orre, L.; Skold, C. M.; Camner, P., Functional and morphological differences between human alveolar and interstitial macrophages. Exp Mol Pathol 2001, 70, (2), 77-82. Gibbings, S. L.; Thomas, S. M.; Atif, S. M.; McCubbrey, A. L.; Desch, A. N.; Danhorn, T.; Leach, S. M.; Bratton, D. L.; Henson, P. M.; Janssen, W. J.; Jakubzick, C. V., Three Unique Interstitial Macrophages in the Murine Lung at Steady State. American journal of respiratory cell and molecular biology 2017, 57, (1), 66-76. Misharin, A. V.; Morales-Nebreda, L.; Mutlu, G. M.; Budinger, G. R.; Perlman, H., Flow cytometric analysis of macrophages and dendritic cell subsets in the mouse lung. American journal of respiratory cell and molecular biology 2013, 49, (4), 503-10. Ferrari-Lacraz, S.; Nicod, L. P.; Chicheportiche, R.; Welgus, H. G.; Dayer, J. M., Human lung tissue macrophages, but not alveolar macrophages, express matrix metalloproteinases after direct contact with activated T lymphocytes. American journal of respiratory cell and molecular biology 2001, 24, (4), 442-51. Kawano, H.; Kayama, H.; Nakama, T.; Hashimoto, T.; Umemoto, E.; Takeda, K., IL-10-producing lung interstitial macrophages prevent neutrophilic asthma. Int Immunol 2016, 28, (10), 489-501. Franke-Ullmann, G.; Pfortner, C.; Walter, P.; Steinmuller, C.; Lohmann-Matthes, M. L.; Kobzik, L., Characterization of murine lung interstitial macrophages in comparison with alveolar macrophages in vitro. Journal of immunology (Baltimore, Md. : 1950) 1996, 157, (7), 3097-104. Wizemann, T. M.; Laskin, D. L., Enhanced phagocytosis, chemotaxis, and production of reactive oxygen intermediates by interstitial lung macrophages following acute endotoxemia. American journal of respiratory cell and molecular biology 1994, 11, (3), 358-65. Toussaint, M.; Fievez, L.; Drion, P. V.; Cataldo, D.; Bureau, F.; Lekeux, P.; Desmet, C. J., Myeloid hypoxia-inducible factor 1alpha prevents airway allergy in mice through macrophage-mediated immunoregulation. Mucosal Immunol 2013, 6, (3), 485-97. Bedoret, D.; Wallemacq, H.; Marichal, T.; Desmet, C.; Quesada Calvo, F.; Henry, E.; Closset, R.; Dewals, B.; Thielen, C.; Gustin, P.; de Leval, L.; Van Rooijen, N.; Le Moine, A.; Vanderplasschen, A.; Cataldo, D.; Drion, P. V.; Moser, M.; Lekeux, P.; Bureau, F., Lung interstitial macrophages alter dendritic cell functions to prevent airway allergy in mice. J Clin Invest 2009, 119, (12), 3723-38. Johansson, A.; Lundborg, M.; Skold, C. M.; Lundahl, J.; Tornling, G.; Eklund, A.; Camner, P., Functional, morphological, and phenotypical differences between rat alveolar and interstitial macrophages. American journal of respiratory cell and molecular biology 1997, 16, (5), 582-8. Lehnert, B. E.; Valdez, Y. E.; Holland, L. M., Pulmonary macrophages: alveolar and interstitial populations. Exp Lung Res 1985, 9, (3-4), 177-90. Sabatel, C.; Radermecker, C.; Fievez, L.; Paulissen, G.; Chakarov, S.; Fernandes, C.; Olivier, S.; Toussaint, M.; Pirottin, D.; Xiao, X.; Quatresooz, P.; Sirard, J. C.; Cataldo, D.; Gillet, L.; Bouabe, H.; Desmet, C. J.; Ginhoux, F.; Marichal, T.; Bureau, F., Exposure to Bacterial CpG DNA Protects from Airway Allergic Inflammation by Expanding Regulatory Lung Interstitial Macrophages. Immunity 2017, 46, (3), 457-473. Cai, Y.; Sugimoto, C.; Arainga, M.; Alvarez, X.; Didier, E. S.; Kuroda, M. J., In vivo characterization of alveolar and interstitial lung macrophages in rhesus macaques: implications for understanding lung disease in humans. Journal of immunology (Baltimore, Md. : 1950) 2014, 192, (6), 2821-9. Bilyk, N.; Mackenzie, J. S.; Papadimitriou, J. M.; Holt, P. G., Functional studies on macrophage populations in the airways and the lung wall of SPF mice in the steady-state and during respiratory virus infection. Immunology 1988, 65, (3), 417-25. Holt, P. G.; Warner, L. A.; Papadimitriou, J. M., Alveolar macrophages: functional heterogeneity within macrophage populations from rat lung. Aust J Exp Biol Med Sci 1982, 60, (6), 607-18. Murphy, J.; Summer, R.; Wilson, A. A.; Kotton, D. N.; Fine, A., The prolonged life-span of alveolar macrophages. American journal of respiratory cell and molecular biology 2008, 38, (4), 380-5. Maus, U. A.; Janzen, S.; Wall, G.; Srivastava, M.; Blackwell, T. S.; Christman, J. W.; Seeger, W.; Welte, T.; Lohmeyer, J., Resident alveolar macrophages are replaced by recruited monocytes in response to endotoxin-induced lung inflammation. American journal of respiratory cell and molecular biology 2006, 35, (2), 227-35. Nakata, K.; Gotoh, H.; Watanabe, J.; Uetake, T.; Komuro, I.; Yuasa, K.; Watanabe, S.; Ieki, R.; Sakamaki, H.; Akiyama, H.; Kudoh, S.; Naitoh, M.; Satoh, H.; Shimada, K., Augmented proliferation of human alveolar macrophages after allogeneic bone marrow transplantation. Blood 1999, 93, (2), 667-73. Landsman, L.; Jung, S., Lung macrophages serve as obligatory intermediate between blood monocytes and alveolar macrophages. Journal of immunology (Baltimore, Md. : 1950) 2007, 179, (6), 3488-94.

Page 3 Line 33: what are the differences in phenotypic markers and functional properties? Please elaborate here.

Response:

Phenotypic markers are static biomarkers (mostly surface markers) while functional properties are specific functions such as chemotaxis, phagocytosis, production of mediators etc. The inflammatory milieu transform the monocytes to be more “macrophages-like” and lead them to secrete the cytokines and chemokines characteristic of macrophages. So, we have changed it as follow.

Page 4, line 17

Before

with phenotypic markers and functional properties

After

with the cytokines and chemokines

Sometimes sections seem out of place: F.Ex.: Page 4 Line 33-38 describing the differences in transcriptional profiles under the heading of chemotaxis. Maybe this could be incorporated into the general section in the beginning of the manuscript?

Response:

We sincerely apologize our description. Without these sentences, this section may be easier to understand. So, have removed it as follow.

Page 5, line 40-43

Recent studies have highlighted that macrophage activation is associated with profound transcriptional reprogramming. Transcriptome-based network analysis has revealed that airspace macrophages isolated from smokers and patients with COPD demonstrate a loss of inflammatory signatures [28].

This review mostly focuses on how LM are affected by and affect other inflammatory cells. A section should also be included on how LM can influence the function/viability of structural cells involved in the pathogenesis of COPD. A summary figure of the different functional characteristics between COPD LM and healthy LM is needed (phenotype, marker, secretion of chemokines or other inflammatory mediators, cell surface marker expression, apoptosis, phagocytosis etc) including the corresponding references. (if there is space limitations, i would rather delete figure 1 as this can be found in almost every review about macrophages and not too much detail is paid on the different phenotypes in this review anyway).

Response: We agree with the review that figure 1 reflect the current paradigm regarding LM but we think it is important to show in a review like this specially for readers that do not follow this field closely. Because LM functional qualities and surface-marker/phenotypes are still an evolving field of research, we have made the figure to reflect current knowledge with references (Figure 3). In addition, to make it easier to understand this figure, I put information about distribution of M1M2 macrophages in COPD in the figure and the text.  

Figure 3. Lung Macrophage (LM) kinetics and phenotypes in chronic obstructive pulmonary disease (COPD): 1. Blood monocytes-derived macrophages (MDMs) migrate into small airways and airspaces and this migration is activated by chemokines such as CCL2 or monocyte chemoattractant protein-1 (MIP-1) [69], CXCL9, CXCL10, CXCL11, and CCL5 (RANTES) [66]. 2. Interstitial macrophage (IM). MDMs are recruited to the lung interstitium and differentiate into IMs [49]. 3. Non-polarized macrophages. IMs recruited  into both small airways and larger airspaces are initially non-polarized and dependent on the micro-environment will polarized to either M1 macrophages or M2 macrophages [12]. 4. M1 macrophages. They secrete pro-inflammatory cytokines such as IL-6, IL-1β and TNF-α [76-79] which could damage lung tissues if secreted in excess. In COPD, phagocytosis and efferocytosis of microorganisms of M1 macrophages are impaired [24, 106, 108-112] that may lead to bacterial colonization. 5. M2 macrophages or alternative acitivated macrophages are involved in resolution of inflammation and tissue repair. In COPD they  produce MMPs (MMP-2, 9, and 12) [52, 57] and cathepsins (cathepsins K, L, and S) [58] as proteases and release significantly less of the tissue inhibitor of metalloproteinase (TIMP)-1 as a anti-elastolytic molecule [52, 87], which could induce tissues damage and amplify the inflammatory response contributing to tissue destruction. Phagocytosis and efferocytosis of cell debris and apoptotic cells are also impaired [105, 113-118]. These apoptotic cells which are not eliminated undergo secondary necrosis with further release of inflammatory mediators that decrease efferocytosis, causing a vicious cycle [124]. The chronic inflammatory process further increases during acute exacerbations and is associated with a poor long-term outcome [5].

Page 2. Line7-9

Before

Furthermore, in the inflammatory milieu in the lung, LM differentiate predominantly into either the classically activated M1 phenotype or the alternatively activated M2 phenotype. The M1 phenotype is generally…

After

Furthermore, in the inflammatory milieu in the lung, in non-polarized macrophages widely distributed in both small airways and airspaces, some migrate to small airways and M1 macrophages are formed, and others migrate to airspaces and M2 macrophages are formed [12]. The M1 phenotype is generally…

Eapen, M. S.; Hansbro, P. M.; McAlinden, K.; Kim, R. Y.; Ward, C.; Hackett, T. L.; Walters, E. H.; Sohal, S. S., Abnormal M1/M2 macrophage phenotype profiles in the small airway wall and lumen in smokers and chronic obstructive pulmonary disease (COPD). Sci Rep 2017, 7, (1), 13392.

(6) Page 1 Line 35: characterized (not characterize)

Page 1 Line 13: increased (not increase)

Legend Figure 2: spelling mistake on apoptotic (apoptoric)

Page 7 Line 37: spelling mistake on apoptotic (apoptostic)

Response:

We sincerely apologize our description. We have changed them.

We apologize that some sentences share high similarity with published papers. So, we have changed the sentences as follow. 

Page 1, line 9-11

Before

Chronic obstructive pulmonary disease (COPD) is caused by the chronic inhalation of toxic particles and gases that initiates a persistent innate and adaptive immune response in the lung.

After

Chronic obstructive pulmonary disease (COPD) is caused by the chronic exposure of the lungs to toxic particles and gases. These exposures initiate a persistent innate and adaptive immune inflammatory response in the airways and lung tissues.

Page 1, line 21-22

Before

to alter or interfere with damaging immune pathways to slow relentless progression of lung tissue destruction thereby improve both morbidity and mortality associated with COPD.

After

to improve impaired immune system, prevent progression of lung tissue destruction, and improve both morbidity and mortality related to COPD.

Page 1, line 27-29

Before

both the large and small airways as well as the lung parenchyma. It is currently the 4th leading cause of chronic morbidity and mortality in the United States and of all chronic diseases, COPD is one where mortality rates are still rising [1]. COPD is caused by the chronic inhalation of toxic particles and gases,

After

the lung parenchyma and the airways. COPD is the 4th leading cause of death in the United States and mortality rates of COPD are still rising compared to other chronic diseases [1]. COPD is caused by the continuing inhalation of toxic substances,

Page 1, line 32-35

Before

These exposures initiate a persistent innate and subsequently an adaptive immune response in the lung. This immune response is associated with an aberrant tissue repair and a remodeling process that results in chronic inflammatory process characterize by an excess production of mucus in the central airways, and permanent destruction and fibrosis of the smaller conducting airways and destruction of the gas exchanging surface in the peripheral lung [4].

After

These exposures initiate a persistent innate and subsequently an adaptive immune response. This response induces an impaired tissue repair and a remodeling process characterized by an overproduction of mucus in the central airways, destruction and fibrosis of the small airways, and destruction of the lung parenchyma [4].

Page 2, line 28-32

Before

Airspace macrophages arise from these embryonic precursors that occupy the luminal niche after the lungs expand at birth. The transepithelial migration and engraftment of these precursors into the airway space are dependent on L-plastin expression [21], and their differentiation into airspace macrophages is critically regulated by granulocyte macrophage-colony stimulating factor (GM-CSF) [22].

After

Airspace macrophages derive from the embryonic precursors that reside the luminal niche after expansion of the lungs at birth. Both the migration through epithelial cells and engraftment of the precursors into the airway are dependent on L-plastin expression [22], and their differentiation into macrophages in the air spaces, is controlled by the expression of granulocyte macrophage-colony stimulating factor (GM-CSF) [23].

Page 2, line 40-43

Before

The resident macrophage pool is then rapidly support by recruited MDMs precursor cells that engraft into the airway space by transepithelial migration and differentiate into airspace macrophages. Both resident and recruited macrophage populations are tightly regulated predominantly

After

The resident macrophage pool is then replaced by recruited MDMs precursor cells that recruited into the airway space via transepithelial migration and differentiate into airspace macrophages. The balance between the resident and recruited macrophage are tightly regulated

Page 2, line 45-46

Before

modify gene transcriptional programs in LM, influencing their phenotype and function.

After

alter the phenotype and function of LM through modifying their gene transcription

Removed the sentence below to overlap the content in “2. Lung Macrophages (LMs) in Chronic Obstructive Pulmonary Disease (COPD)”.

Page 3, line 15-17

Current thought support a M1, M2 differentiation where M1 describes a more pro-inflammatory phenotype associated with killing of intracellular pathogens and M2 providing defense against extracellular parasites and involved in tissue repair.

Page 2, line 50- Page 3, line 3

Before

The “pro-healing” or alternative activated M2 macrophages are characterized by the expression of distinct surface receptors such as mannose receptor (CD206), biosynthetic enzymes such as arachidonate 12,15 lipoxygenases, and other proteins such as chitinases and chitinase-like proteins and matrix metalloproteinase (MMP)-12 [25].

After

M2 macrophages are characterized by the expression of distinguishable surface marker such as mannose receptor (CD206), biosynthetic enzymes such as arachidonate 12,15 lipoxygenases, and other proteins such as chitinases and chitinase-like proteins and matrix metalloproteinase (MMP)-12 [26].

Page 3, line 6-9

Before

subdivided into M2a–d (M2a activation in response to interleukin IL-4 & IL-13, M2b activated by immune complexes and lipopolysaccharides (LPS), M2c activated by glucocorticoids and transforming growth factor beta (TGF-β), and M2d activation in response to IL-6 and adenosines [26, 27]). Therefore the lung micro-environment is capable of shaping LM differentiation and macrophages can be reprogrammed when transferred into a new microenvironment

After

subdivided into M2a–d (M2a promoted by interleukin IL-4 and IL-13, M2b by immune complexes and lipopolysaccharides (LPS), M2c by glucocorticoids and transforming growth factor beta (TGF-β), and M2d by IL-6 and adenosines [27, 28]). Therefore the micro-environment in the lung has ability to shape and reshape LM differentiation dependent on their microenvironment

Page 5 line 25-28

Before

in vivo in models of human and rodents [54-57], have shown that LM are more potent in producing pro-inflammatory cytokines compared to airway epithelial cells [42, 43]. These mediators from LM play a critical role in promoting and regulating the local inflammatory response in the lung following these exposures. Furthermore, LM and airway epithelial cells take advantage of their close proximity to interact in a “paracrine” fashion to amplify and regulate the local inflammatory response in the lung

After

in vivo models in human and rodents [76-79], have shown that LM secrete more   pro-inflammatory cytokines than airway epithelial cells [64, 65]. In addition, LM and airway epithelial cells coordinate the local inflammatory reaction due to their close proximity.

Page 5, line 32-38

Before

These mediators such as macrophage colony-stimulating factor (M-CSF), GM-CSF, IL-6 and others [61], also stimulates the turnover of monocytes in the bone marrow and accelerates their release into the circulation [62]. IL-6 is a strong activator of monocytic colony formation in human hematopoietic progenitor cells in combination with GM-CSF [63]. Mediators such as GM-CSF, IL-6, IL-1β and TNF-α, induce the production of monocytic chemoattractants such as MCP-1, which is a major contributor to the recruitment of peripheral blood monocytes into the alveolar compartment [64]. The stimulation of the bone marrow is proportional to the extent of PM phagocytosis by AM underlining the importance of mediators produced by the resident macrophages in the marrow response to lung inflammation [62]. Whether these newly recruited macrophages are functionally competent to effectively clear toxins, particles and microbes from the air spaces needs to be determined in future studies

After

The turnover of monocytes in the bone marrow is stimulated by macrophage colony-stimulating factor (M-CSF), GM-CSF, IL-6 and also induce their release into the circulation [83][84]. IL-6 activates formation of monocytic colony in hematopoietic progenitor cells in concert with GM-CSF [85]. Mediators such as IL-1β, IL-6, GM-CSF, and TNF-α induce the production of MCP-1 which is a major contributor to the recruitment and replenishment of peripheral blood monocytes into the alveolar space [86]. The stimulation of the bone marrow is related to the ability of PM phagocytosis by AM [84].

Page 5, line 48-50

Before

are dependent on several factors including the nature of the exposure (toxin or particle type, acute or chronic exposure) as well as the host factors such as resident macrophage phenotype and macrophage–epithelium interaction. Further studies are clearly needed to establish the relative importance of all these factors in the pathogenesis of COPD.

After

depend on inhaled particles/pathogens as well as host factors such as the innate and acquired immune responses in the lung. Further studies are needed under considering these factors to clarify the pathogenesis of COPD.

Page 6, line 5-8

Before

and include the fluid-phase uptake mechanism of macropinocytosis [76-78]. Professional phagocytes consist of macrophages and immature dendritic cells (DCs) resident in multiple tissues, tissue-infiltrating monocytes, neutrophils, and eosinophils.

After

and include the uptake of solutes via macropinocytosis [98-100]. Professional phagocytes are composed of macrophages, immature dendritic cells (DCs), tissue-infiltrating monocytes, neutrophils, and eosinophils.

Page 6, line 46- Page7, line1

Before

The transcription factor nuclear factor erythroid 2-related factor 2 (NRF2; also known as NFE2L2) plays an important part in the regulation of antioxidant and cytoprotective genes in response to oxidative stress. In addition, mitochondria release cytochrome C (Cyt C) in response to physical and chemical stress which combined with apoptotic protease activating factor-1 (Apaf-1) and procaspase-9 to form the apoptosome complex [109-111], resulting in the activation of caspase-9 and subsequently caspase-3 which is the final executor of apoptosis [112].

After

The transcription factor nuclear factor erythroid 2-related factor 2 (NRF2; also known as NFE2L2) is important to regulate antioxidant and cytoprotective genes stimulated by oxidative stress. Mitochondria release cytochrome C (Cyt C) induced by physical and chemical stress combined with apoptotic protease activating factor-1 (Apaf-1) and procaspase-9 leading to form the apoptosome complex [131-133], resulting in the activation of caspase-9 and caspase-3 as the final executor of apoptosis [134].

Page 7, line 30-34

Before

The apoptosis induced in these models were associated with increased oxidative stress, Bax protein accumulation, mitochondrial dysfunction, mitochondrial cytochrome c release, but was independent of p53, Fas, and caspase activation [120]. Dose response studies showed that CS extract (CSE) induced apoptosis at lower concentrations (10 to 25%) but cell dead and necrosis at higher concentrations (50 to 100%) [121],

After

The apoptosis induced in these models were related to increase of oxidative stress, Bax protein accumulation, mitochondrial dysfunction, and mitochondrial cytochrome c release, but were unrelated to p53, Fas, and caspase activation [142]. CS extract (CSE) led to apoptosis at lower density (10 to 25%) but cell dead and necrosis at higher density (50 to 100%) [143],

Page 9, line 25-Page10, line 3

Before

Current treatment strategies for COPD consists of bronchodilation, anti-inflammatories and anti-infection therapeutic agents. These treatments have been shown to improve COPD symptoms and reduce exacerbations but currently there is very little evidence that any of these approaches reverse airway remodeling and consequently irreversible obstructive airflow and a decline in forced expiratory volume in 1 second (FEV1) over time. Cessation of smoking is currently the only intervention that has convincingly been shown to reduce the decline in FEV1 over time.

After

Bronchodilation, anti-inflammatories and anti-infection drugs are generally used as treatment strategies for COPD. These interventions have been shown to improve COPD symptoms and prevent exacerbations, but there is very little evidence that these approaches improve airway remodeling, irreversible obstructive airflow, and a decline in forced expiratory volume in 1 second (FEV1) over time. Smoking cessation is the only intervention that has convincingly been shown to prevent the decline in FEV1 over time.

Page 10, line 10-Page10-14

Before

Several studies including a recent meta-analysis of the use of macrolides in reducing COPD exacerbations, showed a significant reduction (OR 0.28 CI 0.12-0.68) with low dose of macrolides continuously over the full year of the study [150]. This provides evidence of clinical benefit using macrolides as immune modulators in COPD. Macrolides, are known to have wide immune-modulatory effects on airway inflammation. They suppress activator protein 1 (AP1) and the NF-κB mediated cascade resulting in reduced production of IL-1, IL-6, IL-8, TNF-α predominantly made by LM in COPD patients

After

The recent meta-analysis showed that low dose macrolides therapy significantly reduced exacerbations (OR 0.28 CI 0.12-0.68) [172]. Macrolides are known to have immune-modulatory effects on chronic airway inflammation. They can suppress activator protein 1 (AP1) and the NF-κB mediated cascade leading to reduce production of pro-inflammatory mediators such as IL-1, IL-6, IL-8, TNF-α predominantly made by LM in COPD patients

Page 10, line 16-18

Before

thus reducing further inflammatory responses related to cell necrosis; alter macrophage phenotype to improve bacterial clearance, and enhance macrophage cytocidal activity in COPD [99, 100, 151]

After

thus suppressing excessive inflammatory responses related to cell necrosis; change macrophage phenotype to improve bacterial clearance, and raise macrophage cytocidal activity in COPD [121, 122, 173]

Page 10, line 34-36

Before

The collectin family mainly includes mannose binding lectin (MBL) and surfactant proteins A (SP-A), and D (SP-D). Collectin consisits of collagen-like amino (N)-terminal regions and C-type (calcium dependent) carbohydrate-recognition domains (CRDs).

After

The collectin family consists of the mannose binding lectin (MBL), surfactant proteins A (SP-A), and D (SP-D), with a collagen-like amino (N)-terminal regions and C-type (calcium dependent) carbohydrate-recognition domains (CRDs).

Page 11, line 16-18

Before

These findings indicate that statins strengthen efferocytosis in vitro and in vivo mediated by RhoA inhibition [176]. CS and other oxidant stresses also reduce efferocytosis by activating RhoA [181]

After

These findings show that statins enhance efferocytosis via RhoA inhibition in vitro and in vivo [198]. CS and other oxidant stressors, suppress efferocytosis via activating RhoA [203]

Page 11, line 35-37

Before

Selective PDE4 inhibitors have broad spectrum anti-inflammatory effects such as inhibition of cell trafficking, cytokine and chemokine release from inflammatory cells, such as macrophages. The second generation of PDE4 inhibitors such as roflumilast clinical benefit

After

Selective PDE4 inhibitors are broad spectrum anti-inflammatory drugs inhibiting cell trafficking, produce of cytokine and chemokine from macrophages. The second generation of PDE4 inhibitors such as roflumilast clinical benefit

Reviewer 2 Report

This article by Akata and van Eeden is a very thorough and profound review on the current literature on lung macrophages in COPD.

The authors have covered the majority of available literature.

There are only a few points which would need further investigation:

The general part of the review occasionally appears unstructured and sometimes redundant. I would suggest to carefully scan Page 1-3 for redundant passages and rather explained it in a structured manner once than in short passages multiple times. A section about the different types of lung macrophages according to their localization would add a new perspective to this review: alveolar vs interstitial. How are the different?What are the differences? Functions? Phagocytic capacity etc? A small section about their different characteristics would be nice. How are they affected by COPD? Page 3 Line 33: what are the differences in phenotypic markers and functional properties? Please elaborate here. Sometimes sections seem out of place: F.Ex.: Page 4 Line 33-38 describing the differences in transcriptional profiles under the heading of chemotaxis. Maybe this could be incorporated into the general section in the beginning of the manuscript? This review mostly focuses on how LM are affected by and affect other inflammatory cells. A section should also be included on how LM can influence the function/viability of structural cells involved in the pathogenesis of COPD. A summary figure of the different functional characteristics between COPD LM and healthy LM is needed (phenotype, marker, secretion of chemokines or other inflammatory mediators, cell surface marker expression, apoptosis, phagocytosis etc) including the corresponding references. (if there is space limitations, i would rather delete figure 1 as this can be found in almost every review about macrophages and not too much detail is paid on the different phenotypes in this review anyway).

Page 1 Line 35: characterized (not characterize)

Page 1 Line 13: increased (not increase)

Legend Figure 2: spelling mistake on apoptotic (apoptoric)

Page 7 Line 37: spelling mistake on apoptotic (apoptostic)

Author Response

Responses to the reviewer’s comments (Reviewer 2):

Reviewer #2:

We really appreciate your efforts to review our manuscript and your important review comments. According to your comments, we have revised our manuscript as follows.

Minor comments

The general part of the review occasionally appears unstructured and sometimes redundant. I would suggest to carefully scan Page 1-3 for redundant passages and rather explained it in a structured manner once than in short passages multiple times.

Response:

We have removed redundant and repetitive sections as requested (see MS)

A section about the different types of lung macrophages according to their localization would add a new perspective to this review: alveolar vs interstitial. How are the different? What are the differences? Functions? Phagocytic capacity etc? A small section about their different characteristics would be nice. How are they affected by COPD?

Response:

Thank you very much for your important comments. We also think it is important of the difference between alveolar macrophages and interstitial macrophages. Therefore, we added the paragraph as below.

Page 3, line 22-Page 4, line 10

Alveolar Macrophage (AM) vs Interstitial macrophage (IM)

Lung macrophages mainly consist of airspace (AM) and interstitial macrophages (IM) based on their anatomical location. There are less informative studies using IM because fresh lung tissue is needed to extract these macrophages for functional studies. Subsequent culture of these extracted macrophages could also alter their functional properties. IMs are smaller and morphologically more similar blood monocytes than AM [30-32]. IMs also have a higher nuclear/cytoplasm ratio and their cytosol includes more vacuoles [33] and are more heterogeneous in shape compared to AM [30, 31].

IM have a lower phagocytic activity of Saccharomyces cerevisiae [31] but their phagocytic activity of latex beads is similar to AM [29], which means lower Fcϒ-dependent phagocytic activity in IM than AM. IM expressed more MHC-II (HLA-DR) to function as antigen-presenting cells [30, 34]. In the steady-state, IMs secrete pro-inflammatory cytokines (IL-1, IL-6 [30], and TNF-α [35-37]) and anti-inflammatory cytokine (IL-10 [38-41]). The amount of IL-10 produce by IMs increases in response to the stimuli such as LPS [39], DNA containing non-methylated CpG motifs (CpG-DNA) [42] or extracts of house dust mite (HDM) [38] and expression of TNF-α increased in IMs but not in AMs in response of IFN-ϒ and LPS [43]. Expression of matrix metalloproteinases (MMP) in IM was higher than in AM [34]. Blood monocytes can transition into IMs that can be recruited into airspaces and transition to AMs via a process of maturation [44, 45]. The turnover rate of IMs is shorter in steady-state than that of AMs, predominantly regulated by apoptosis [43] resulting in AMs living longer than IMs [46-48]. In addition, lung inflammation recruited monocyte via lung tissue into airspaces where they differentiation into AM [49]. Furthermore, macrophages with AM phenotype have been identified in the lung interstitium [50] and vice versa [42] suggesting that these two populations of macrophages replenish each other dependent on need. This plasticity of LM complicate studies on LM [49]. Further in in vivo studies are needed to better clarify phenotypic and functional characteristics of LM and their potential role in the pathogenesis of COPD. 

Xue, J.; Schmidt, S. V.; Sander, J.; Draffehn, A.; Krebs, W.; Quester, I.; De Nardo, D.; Gohel, T. D.; Emde, M.; Schmidleithner, L.; Ganesan, H.; Nino-Castro, A.; Mallmann, M. R.; Labzin, L.; Theis, H.; Kraut, M.; Beyer, M.; Latz, E.; Freeman, T. C.; Ulas, T.; Schultze, J. L., Transcriptome-based network analysis reveals a spectrum model of human macrophage activation. Immunity 2014, 40, (2), 274-88. Hoppstadter, J.; Diesel, B.; Zarbock, R.; Breinig, T.; Monz, D.; Koch, M.; Meyerhans, A.; Gortner, L.; Lehr, C. M.; Huwer, H.; Kiemer, A. K., Differential cell reaction upon Toll-like receptor 4 and 9 activation in human alveolar and lung interstitial macrophages. Respiratory research 2010, 11, 124. Fathi, M.; Johansson, A.; Lundborg, M.; Orre, L.; Skold, C. M.; Camner, P., Functional and morphological differences between human alveolar and interstitial macrophages. Exp Mol Pathol 2001, 70, (2), 77-82. Gibbings, S. L.; Thomas, S. M.; Atif, S. M.; McCubbrey, A. L.; Desch, A. N.; Danhorn, T.; Leach, S. M.; Bratton, D. L.; Henson, P. M.; Janssen, W. J.; Jakubzick, C. V., Three Unique Interstitial Macrophages in the Murine Lung at Steady State. American journal of respiratory cell and molecular biology 2017, 57, (1), 66-76. Misharin, A. V.; Morales-Nebreda, L.; Mutlu, G. M.; Budinger, G. R.; Perlman, H., Flow cytometric analysis of macrophages and dendritic cell subsets in the mouse lung. American journal of respiratory cell and molecular biology 2013, 49, (4), 503-10. Ferrari-Lacraz, S.; Nicod, L. P.; Chicheportiche, R.; Welgus, H. G.; Dayer, J. M., Human lung tissue macrophages, but not alveolar macrophages, express matrix metalloproteinases after direct contact with activated T lymphocytes. American journal of respiratory cell and molecular biology 2001, 24, (4), 442-51. Kawano, H.; Kayama, H.; Nakama, T.; Hashimoto, T.; Umemoto, E.; Takeda, K., IL-10-producing lung interstitial macrophages prevent neutrophilic asthma. Int Immunol 2016, 28, (10), 489-501. Franke-Ullmann, G.; Pfortner, C.; Walter, P.; Steinmuller, C.; Lohmann-Matthes, M. L.; Kobzik, L., Characterization of murine lung interstitial macrophages in comparison with alveolar macrophages in vitro. Journal of immunology (Baltimore, Md. : 1950) 1996, 157, (7), 3097-104. Wizemann, T. M.; Laskin, D. L., Enhanced phagocytosis, chemotaxis, and production of reactive oxygen intermediates by interstitial lung macrophages following acute endotoxemia. American journal of respiratory cell and molecular biology 1994, 11, (3), 358-65. Toussaint, M.; Fievez, L.; Drion, P. V.; Cataldo, D.; Bureau, F.; Lekeux, P.; Desmet, C. J., Myeloid hypoxia-inducible factor 1alpha prevents airway allergy in mice through macrophage-mediated immunoregulation. Mucosal Immunol 2013, 6, (3), 485-97. Bedoret, D.; Wallemacq, H.; Marichal, T.; Desmet, C.; Quesada Calvo, F.; Henry, E.; Closset, R.; Dewals, B.; Thielen, C.; Gustin, P.; de Leval, L.; Van Rooijen, N.; Le Moine, A.; Vanderplasschen, A.; Cataldo, D.; Drion, P. V.; Moser, M.; Lekeux, P.; Bureau, F., Lung interstitial macrophages alter dendritic cell functions to prevent airway allergy in mice. J Clin Invest 2009, 119, (12), 3723-38. Johansson, A.; Lundborg, M.; Skold, C. M.; Lundahl, J.; Tornling, G.; Eklund, A.; Camner, P., Functional, morphological, and phenotypical differences between rat alveolar and interstitial macrophages. American journal of respiratory cell and molecular biology 1997, 16, (5), 582-8. Lehnert, B. E.; Valdez, Y. E.; Holland, L. M., Pulmonary macrophages: alveolar and interstitial populations. Exp Lung Res 1985, 9, (3-4), 177-90. Sabatel, C.; Radermecker, C.; Fievez, L.; Paulissen, G.; Chakarov, S.; Fernandes, C.; Olivier, S.; Toussaint, M.; Pirottin, D.; Xiao, X.; Quatresooz, P.; Sirard, J. C.; Cataldo, D.; Gillet, L.; Bouabe, H.; Desmet, C. J.; Ginhoux, F.; Marichal, T.; Bureau, F., Exposure to Bacterial CpG DNA Protects from Airway Allergic Inflammation by Expanding Regulatory Lung Interstitial Macrophages. Immunity 2017, 46, (3), 457-473. Cai, Y.; Sugimoto, C.; Arainga, M.; Alvarez, X.; Didier, E. S.; Kuroda, M. J., In vivo characterization of alveolar and interstitial lung macrophages in rhesus macaques: implications for understanding lung disease in humans. Journal of immunology (Baltimore, Md. : 1950) 2014, 192, (6), 2821-9. Bilyk, N.; Mackenzie, J. S.; Papadimitriou, J. M.; Holt, P. G., Functional studies on macrophage populations in the airways and the lung wall of SPF mice in the steady-state and during respiratory virus infection. Immunology 1988, 65, (3), 417-25. Holt, P. G.; Warner, L. A.; Papadimitriou, J. M., Alveolar macrophages: functional heterogeneity within macrophage populations from rat lung. Aust J Exp Biol Med Sci 1982, 60, (6), 607-18. Murphy, J.; Summer, R.; Wilson, A. A.; Kotton, D. N.; Fine, A., The prolonged life-span of alveolar macrophages. American journal of respiratory cell and molecular biology 2008, 38, (4), 380-5. Maus, U. A.; Janzen, S.; Wall, G.; Srivastava, M.; Blackwell, T. S.; Christman, J. W.; Seeger, W.; Welte, T.; Lohmeyer, J., Resident alveolar macrophages are replaced by recruited monocytes in response to endotoxin-induced lung inflammation. American journal of respiratory cell and molecular biology 2006, 35, (2), 227-35. Nakata, K.; Gotoh, H.; Watanabe, J.; Uetake, T.; Komuro, I.; Yuasa, K.; Watanabe, S.; Ieki, R.; Sakamaki, H.; Akiyama, H.; Kudoh, S.; Naitoh, M.; Satoh, H.; Shimada, K., Augmented proliferation of human alveolar macrophages after allogeneic bone marrow transplantation. Blood 1999, 93, (2), 667-73. Landsman, L.; Jung, S., Lung macrophages serve as obligatory intermediate between blood monocytes and alveolar macrophages. Journal of immunology (Baltimore, Md. : 1950) 2007, 179, (6), 3488-94.

Page 3 Line 33: what are the differences in phenotypic markers and functional properties? Please elaborate here.

Response:

Phenotypic markers are static biomarkers (mostly surface markers) while functional properties are specific functions such as chemotaxis, phagocytosis, production of mediators etc. The inflammatory milieu transform the monocytes to be more “macrophages-like” and lead them to secrete the cytokines and chemokines characteristic of macrophages. So, we have changed it as follow.

Page 4, line 17

Before

with phenotypic markers and functional properties

After

with the cytokines and chemokines

Sometimes sections seem out of place: F.Ex.: Page 4 Line 33-38 describing the differences in transcriptional profiles under the heading of chemotaxis. Maybe this could be incorporated into the general section in the beginning of the manuscript?

Response:

We sincerely apologize our description. Without these sentences, this section may be easier to understand. So, have removed it as follow.

Page 5, line 40-43

Recent studies have highlighted that macrophage activation is associated with profound transcriptional reprogramming. Transcriptome-based network analysis has revealed that airspace macrophages isolated from smokers and patients with COPD demonstrate a loss of inflammatory signatures [28].

This review mostly focuses on how LM are affected by and affect other inflammatory cells. A section should also be included on how LM can influence the function/viability of structural cells involved in the pathogenesis of COPD. A summary figure of the different functional characteristics between COPD LM and healthy LM is needed (phenotype, marker, secretion of chemokines or other inflammatory mediators, cell surface marker expression, apoptosis, phagocytosis etc) including the corresponding references. (if there is space limitations, i would rather delete figure 1 as this can be found in almost every review about macrophages and not too much detail is paid on the different phenotypes in this review anyway).

Response: We agree with the review that figure 1 reflect the current paradigm regarding LM but we think it is important to show in a review like this specially for readers that do not follow this field closely. Because LM functional qualities and surface-marker/phenotypes are still an evolving field of research, we have made the figure to reflect current knowledge with references (Figure 3). In addition, to make it easier to understand this figure, I put information about distribution of M1M2 macrophages in COPD in the figure and the text.  

Figure 3. Lung Macrophage (LM) kinetics and phenotypes in chronic obstructive pulmonary disease (COPD): 1. Blood monocytes-derived macrophages (MDMs) migrate into small airways and airspaces and this migration is activated by chemokines such as CCL2 or monocyte chemoattractant protein-1 (MIP-1) [69], CXCL9, CXCL10, CXCL11, and CCL5 (RANTES) [66]. 2. Interstitial macrophage (IM). MDMs are recruited to the lung interstitium and differentiate into IMs [49]. 3. Non-polarized macrophages. IMs recruited  into both small airways and larger airspaces are initially non-polarized and dependent on the micro-environment will polarized to either M1 macrophages or M2 macrophages [12]. 4. M1 macrophages. They secrete pro-inflammatory cytokines such as IL-6, IL-1β and TNF-α [76-79] which could damage lung tissues if secreted in excess. In COPD, phagocytosis and efferocytosis of microorganisms of M1 macrophages are impaired [24, 106, 108-112] that may lead to bacterial colonization. 5. M2 macrophages or alternative acitivated macrophages are involved in resolution of inflammation and tissue repair. In COPD they  produce MMPs (MMP-2, 9, and 12) [52, 57] and cathepsins (cathepsins K, L, and S) [58] as proteases and release significantly less of the tissue inhibitor of metalloproteinase (TIMP)-1 as a anti-elastolytic molecule [52, 87], which could induce tissues damage and amplify the inflammatory response contributing to tissue destruction. Phagocytosis and efferocytosis of cell debris and apoptotic cells are also impaired [105, 113-118]. These apoptotic cells which are not eliminated undergo secondary necrosis with further release of inflammatory mediators that decrease efferocytosis, causing a vicious cycle [124]. The chronic inflammatory process further increases during acute exacerbations and is associated with a poor long-term outcome [5].

Page 2. Line7-9

Before

Furthermore, in the inflammatory milieu in the lung, LM differentiate predominantly into either the classically activated M1 phenotype or the alternatively activated M2 phenotype. The M1 phenotype is generally…

After

Furthermore, in the inflammatory milieu in the lung, in non-polarized macrophages widely distributed in both small airways and airspaces, some migrate to small airways and M1 macrophages are formed, and others migrate to airspaces and M2 macrophages are formed [12]. The M1 phenotype is generally…

Eapen, M. S.; Hansbro, P. M.; McAlinden, K.; Kim, R. Y.; Ward, C.; Hackett, T. L.; Walters, E. H.; Sohal, S. S., Abnormal M1/M2 macrophage phenotype profiles in the small airway wall and lumen in smokers and chronic obstructive pulmonary disease (COPD). Sci Rep 2017, 7, (1), 13392.

(6) Page 1 Line 35: characterized (not characterize)

Page 1 Line 13: increased (not increase)

Legend Figure 2: spelling mistake on apoptotic (apoptoric)

Page 7 Line 37: spelling mistake on apoptotic (apoptostic)

Response:

We sincerely apologize our description. We have changed them.

Responses to the reviewer’s comments (Reviewer 1):

Reviewer #1:

We really appreciate your efforts to review our manuscript and your important review comments. According to your comments, we have revised our manuscript as follows.

Minor comments

(1) The review is written in an educational manner, thus I think that the authors should explain in better way what does it means E10.5 (in the paragraph Origins of LM) and also that F4/80 is the major macrophage marker.

Response:

Thank you very much for your important comments. We changed the sentence as below.

Page 2, line 20-21

Before

The first wave starts at E10.5 in yolk sac. F4/80 lineage macrophages …

After

The first wave starts at embryonic day 10.5 (E10.5) in yolk sac. F4/80 (which is the major macrophage marker) lineage macrophages…

(2) In the legend of the Figure 2 there are too many parenthesis that make difficult the reading. Consider to put the abbreviations at the end of the paper.

Response:

We sincerely apologize our description in Figure 2. We have changed it as follow.

Figure 2

Before

Figure 2. Apoptosis in lung macrophage is determined by a variety of apoptoric (oxidative stress, B-cell lymphoma protein 2 (Bcl-2)-associated X (Bax), mitochondrial dysfunction (cytochrome c release) and anti-apoptostic (p21CIP1/WAF, B-cell lymphoma-extra large (Bcl-xL), and v-maf avian musculoaponeurotic fibrosarcoma oncogene homolog B (MafB) factors. Exposure to cigarrete smoke and pathogens impact this balance of anti-apoptotic and apoptotic factors. This could contribute to reduced clearance of dead cell and cell debri’s augmenting the chronic inflammatory response in lung tissues of chronic obstructive pulmonary disease subjects.

After

Figure 2. Apoptosis in lung macrophage is determined by a variety of apoptotic (oxidative stress, Bax, and mitochondrial dysfunction such as cytochrome c release) and anti-apoptostic (p21CIP1/WAF, Bcl-xL, and MafB) factors. Exposure to cigarrete smoke and pathogens impact this balance of anti-apoptotic and apoptotic factors. This could contribute to reduced clearance of dead cell and cell debri’s augmenting the chronic inflammatory response in lung tissues of chronic obstructive pulmonary disease subjects.

Add

Page 13, line 2-5

Abbreviations

Bax    B-cell lymphoma protein 2-associated X

Bcl-xL  B-cell lymphoma-extra large

MafB   v-maf avian musculoaponeurotic fibrosarcoma oncogene homolog B

(3) Lane 37, page 8: hyperoxia (95% O2) instead of (95% O2).

Figure 3: the “S” of ROS is under the arrow.

Response:

We sincerely apologize our description. We have changed it as bellow.

Page 8, line 46

Before

hyperoxia (95% O2)

After

hyperoxia (95% O2)

Figure 3

We have changed the figure.

We apologize that some sentences share high similarity with published papers. So, we have changed the sentences as follow. 

Page 1, line 9-11

Before

Chronic obstructive pulmonary disease (COPD) is caused by the chronic inhalation of toxic particles and gases that initiates a persistent innate and adaptive immune response in the lung.

After

Chronic obstructive pulmonary disease (COPD) is caused by the chronic exposure of the lungs to toxic particles and gases. These exposures initiate a persistent innate and adaptive immune inflammatory response in the airways and lung tissues.

Page 1, line 21-22

Before

to alter or interfere with damaging immune pathways to slow relentless progression of lung tissue destruction thereby improve both morbidity and mortality associated with COPD.

After

to improve impaired immune system, prevent progression of lung tissue destruction, and improve both morbidity and mortality related to COPD.

Page 1, line 27-29

Before

both the large and small airways as well as the lung parenchyma. It is currently the 4th leading cause of chronic morbidity and mortality in the United States and of all chronic diseases, COPD is one where mortality rates are still rising [1]. COPD is caused by the chronic inhalation of toxic particles and gases,

After

the lung parenchyma and the airways. COPD is the 4th leading cause of death in the United States and mortality rates of COPD are still rising compared to other chronic diseases [1]. COPD is caused by the continuing inhalation of toxic substances,

Page 1, line 32-35

Before

These exposures initiate a persistent innate and subsequently an adaptive immune response in the lung. This immune response is associated with an aberrant tissue repair and a remodeling process that results in chronic inflammatory process characterize by an excess production of mucus in the central airways, and permanent destruction and fibrosis of the smaller conducting airways and destruction of the gas exchanging surface in the peripheral lung [4].

After

These exposures initiate a persistent innate and subsequently an adaptive immune response. This response induces an impaired tissue repair and a remodeling process characterized by an overproduction of mucus in the central airways, destruction and fibrosis of the small airways, and destruction of the lung parenchyma [4].

Page 2, line 28-32

Before

Airspace macrophages arise from these embryonic precursors that occupy the luminal niche after the lungs expand at birth. The transepithelial migration and engraftment of these precursors into the airway space are dependent on L-plastin expression [21], and their differentiation into airspace macrophages is critically regulated by granulocyte macrophage-colony stimulating factor (GM-CSF) [22].

After

Airspace macrophages derive from the embryonic precursors that reside the luminal niche after expansion of the lungs at birth. Both the migration through epithelial cells and engraftment of the precursors into the airway are dependent on L-plastin expression [22], and their differentiation into macrophages in the air spaces, is controlled by the expression of granulocyte macrophage-colony stimulating factor (GM-CSF) [23].

Page 2, line 40-43

Before

The resident macrophage pool is then rapidly support by recruited MDMs precursor cells that engraft into the airway space by transepithelial migration and differentiate into airspace macrophages. Both resident and recruited macrophage populations are tightly regulated predominantly

After

The resident macrophage pool is then replaced by recruited MDMs precursor cells that recruited into the airway space via transepithelial migration and differentiate into airspace macrophages. The balance between the resident and recruited macrophage are tightly regulated

Page 2, line 45-46

Before

modify gene transcriptional programs in LM, influencing their phenotype and function.

After

alter the phenotype and function of LM through modifying their gene transcription

Removed the sentence below to overlap the content in “2. Lung Macrophages (LMs) in Chronic Obstructive Pulmonary Disease (COPD)”.

Page 3, line 15-17

Current thought support a M1, M2 differentiation where M1 describes a more pro-inflammatory phenotype associated with killing of intracellular pathogens and M2 providing defense against extracellular parasites and involved in tissue repair.

Page 2, line 50- Page 3, line 3

Before

The “pro-healing” or alternative activated M2 macrophages are characterized by the expression of distinct surface receptors such as mannose receptor (CD206), biosynthetic enzymes such as arachidonate 12,15 lipoxygenases, and other proteins such as chitinases and chitinase-like proteins and matrix metalloproteinase (MMP)-12 [25].

After

M2 macrophages are characterized by the expression of distinguishable surface marker such as mannose receptor (CD206), biosynthetic enzymes such as arachidonate 12,15 lipoxygenases, and other proteins such as chitinases and chitinase-like proteins and matrix metalloproteinase (MMP)-12 [26].

Page 3, line 6-9

Before

subdivided into M2a–d (M2a activation in response to interleukin IL-4 & IL-13, M2b activated by immune complexes and lipopolysaccharides (LPS), M2c activated by glucocorticoids and transforming growth factor beta (TGF-β), and M2d activation in response to IL-6 and adenosines [26, 27]). Therefore the lung micro-environment is capable of shaping LM differentiation and macrophages can be reprogrammed when transferred into a new microenvironment

After

subdivided into M2a–d (M2a promoted by interleukin IL-4 and IL-13, M2b by immune complexes and lipopolysaccharides (LPS), M2c by glucocorticoids and transforming growth factor beta (TGF-β), and M2d by IL-6 and adenosines [27, 28]). Therefore the micro-environment in the lung has ability to shape and reshape LM differentiation dependent on their microenvironment

Page 5 line 25-28

Before

in vivo in models of human and rodents [54-57], have shown that LM are more potent in producing pro-inflammatory cytokines compared to airway epithelial cells [42, 43]. These mediators from LM play a critical role in promoting and regulating the local inflammatory response in the lung following these exposures. Furthermore, LM and airway epithelial cells take advantage of their close proximity to interact in a “paracrine” fashion to amplify and regulate the local inflammatory response in the lung

After

in vivo models in human and rodents [76-79], have shown that LM secrete more   pro-inflammatory cytokines than airway epithelial cells [64, 65]. In addition, LM and airway epithelial cells coordinate the local inflammatory reaction due to their close proximity.

Page 5, line 32-38

Before

These mediators such as macrophage colony-stimulating factor (M-CSF), GM-CSF, IL-6 and others [61], also stimulates the turnover of monocytes in the bone marrow and accelerates their release into the circulation [62]. IL-6 is a strong activator of monocytic colony formation in human hematopoietic progenitor cells in combination with GM-CSF [63]. Mediators such as GM-CSF, IL-6, IL-1β and TNF-α, induce the production of monocytic chemoattractants such as MCP-1, which is a major contributor to the recruitment of peripheral blood monocytes into the alveolar compartment [64]. The stimulation of the bone marrow is proportional to the extent of PM phagocytosis by AM underlining the importance of mediators produced by the resident macrophages in the marrow response to lung inflammation [62]. Whether these newly recruited macrophages are functionally competent to effectively clear toxins, particles and microbes from the air spaces needs to be determined in future studies

After

The turnover of monocytes in the bone marrow is stimulated by macrophage colony-stimulating factor (M-CSF), GM-CSF, IL-6 and also induce their release into the circulation [83][84]. IL-6 activates formation of monocytic colony in hematopoietic progenitor cells in concert with GM-CSF [85]. Mediators such as IL-1β, IL-6, GM-CSF, and TNF-α induce the production of MCP-1 which is a major contributor to the recruitment and replenishment of peripheral blood monocytes into the alveolar space [86]. The stimulation of the bone marrow is related to the ability of PM phagocytosis by AM [84].

Page 5, line 48-50

Before

are dependent on several factors including the nature of the exposure (toxin or particle type, acute or chronic exposure) as well as the host factors such as resident macrophage phenotype and macrophage–epithelium interaction. Further studies are clearly needed to establish the relative importance of all these factors in the pathogenesis of COPD.

After

depend on inhaled particles/pathogens as well as host factors such as the innate and acquired immune responses in the lung. Further studies are needed under considering these factors to clarify the pathogenesis of COPD.

Page 6, line 5-8

Before

and include the fluid-phase uptake mechanism of macropinocytosis [76-78]. Professional phagocytes consist of macrophages and immature dendritic cells (DCs) resident in multiple tissues, tissue-infiltrating monocytes, neutrophils, and eosinophils.

After

and include the uptake of solutes via macropinocytosis [98-100]. Professional phagocytes are composed of macrophages, immature dendritic cells (DCs), tissue-infiltrating monocytes, neutrophils, and eosinophils.

Page 6, line 46- Page7, line1

Before

The transcription factor nuclear factor erythroid 2-related factor 2 (NRF2; also known as NFE2L2) plays an important part in the regulation of antioxidant and cytoprotective genes in response to oxidative stress. In addition, mitochondria release cytochrome C (Cyt C) in response to physical and chemical stress which combined with apoptotic protease activating factor-1 (Apaf-1) and procaspase-9 to form the apoptosome complex [109-111], resulting in the activation of caspase-9 and subsequently caspase-3 which is the final executor of apoptosis [112].

After

The transcription factor nuclear factor erythroid 2-related factor 2 (NRF2; also known as NFE2L2) is important to regulate antioxidant and cytoprotective genes stimulated by oxidative stress. Mitochondria release cytochrome C (Cyt C) induced by physical and chemical stress combined with apoptotic protease activating factor-1 (Apaf-1) and procaspase-9 leading to form the apoptosome complex [131-133], resulting in the activation of caspase-9 and caspase-3 as the final executor of apoptosis [134].

Page 7, line 30-34

Before

The apoptosis induced in these models were associated with increased oxidative stress, Bax protein accumulation, mitochondrial dysfunction, mitochondrial cytochrome c release, but was independent of p53, Fas, and caspase activation [120]. Dose response studies showed that CS extract (CSE) induced apoptosis at lower concentrations (10 to 25%) but cell dead and necrosis at higher concentrations (50 to 100%) [121],

After

The apoptosis induced in these models were related to increase of oxidative stress, Bax protein accumulation, mitochondrial dysfunction, and mitochondrial cytochrome c release, but were unrelated to p53, Fas, and caspase activation [142]. CS extract (CSE) led to apoptosis at lower density (10 to 25%) but cell dead and necrosis at higher density (50 to 100%) [143],

Page 9, line 25-Page10, line 3

Before

Current treatment strategies for COPD consists of bronchodilation, anti-inflammatories and anti-infection therapeutic agents. These treatments have been shown to improve COPD symptoms and reduce exacerbations but currently there is very little evidence that any of these approaches reverse airway remodeling and consequently irreversible obstructive airflow and a decline in forced expiratory volume in 1 second (FEV1) over time. Cessation of smoking is currently the only intervention that has convincingly been shown to reduce the decline in FEV1 over time.

After

Bronchodilation, anti-inflammatories and anti-infection drugs are generally used as treatment strategies for COPD. These interventions have been shown to improve COPD symptoms and prevent exacerbations, but there is very little evidence that these approaches improve airway remodeling, irreversible obstructive airflow, and a decline in forced expiratory volume in 1 second (FEV1) over time. Smoking cessation is the only intervention that has convincingly been shown to prevent the decline in FEV1 over time.

Page 10, line 10-Page10-14

Before

Several studies including a recent meta-analysis of the use of macrolides in reducing COPD exacerbations, showed a significant reduction (OR 0.28 CI 0.12-0.68) with low dose of macrolides continuously over the full year of the study [150]. This provides evidence of clinical benefit using macrolides as immune modulators in COPD. Macrolides, are known to have wide immune-modulatory effects on airway inflammation. They suppress activator protein 1 (AP1) and the NF-κB mediated cascade resulting in reduced production of IL-1, IL-6, IL-8, TNF-α predominantly made by LM in COPD patients

After

The recent meta-analysis showed that low dose macrolides therapy significantly reduced exacerbations (OR 0.28 CI 0.12-0.68) [172]. Macrolides are known to have immune-modulatory effects on chronic airway inflammation. They can suppress activator protein 1 (AP1) and the NF-κB mediated cascade leading to reduce production of pro-inflammatory mediators such as IL-1, IL-6, IL-8, TNF-α predominantly made by LM in COPD patients

Page 10, line 16-18

Before

thus reducing further inflammatory responses related to cell necrosis; alter macrophage phenotype to improve bacterial clearance, and enhance macrophage cytocidal activity in COPD [99, 100, 151]

After

thus suppressing excessive inflammatory responses related to cell necrosis; change macrophage phenotype to improve bacterial clearance, and raise macrophage cytocidal activity in COPD [121, 122, 173]

Page 10, line 34-36

Before

The collectin family mainly includes mannose binding lectin (MBL) and surfactant proteins A (SP-A), and D (SP-D). Collectin consisits of collagen-like amino (N)-terminal regions and C-type (calcium dependent) carbohydrate-recognition domains (CRDs).

After

The collectin family consists of the mannose binding lectin (MBL), surfactant proteins A (SP-A), and D (SP-D), with a collagen-like amino (N)-terminal regions and C-type (calcium dependent) carbohydrate-recognition domains (CRDs).

Page 11, line 16-18

Before

These findings indicate that statins strengthen efferocytosis in vitro and in vivo mediated by RhoA inhibition [176]. CS and other oxidant stresses also reduce efferocytosis by activating RhoA [181]

After

These findings show that statins enhance efferocytosis via RhoA inhibition in vitro and in vivo [198]. CS and other oxidant stressors, suppress efferocytosis via activating RhoA [203]

Page 11, line 35-37

Before

Selective PDE4 inhibitors have broad spectrum anti-inflammatory effects such as inhibition of cell trafficking, cytokine and chemokine release from inflammatory cells, such as macrophages. The second generation of PDE4 inhibitors such as roflumilast clinical benefit

After

Selective PDE4 inhibitors are broad spectrum anti-inflammatory drugs inhibiting cell trafficking, produce of cytokine and chemokine from macrophages. The second generation of PDE4 inhibitors such as roflumilast clinical benefit